# EgoEnv: Human-centric environment representations from egocentric video

**Tushar Nagarajan**[2], **Santhosh Kumar Ramakrishnan**[1], **Ruta Desai**[2],
**James Hillis**[2], **Kristen Grauman**[1,2]
[1]University of Texas at Austin, [2]FAIR, Meta

## Abstract

First-person video highlights a camera-wearer's activities *in the context of their persistent environment*. However, current video understanding approaches reason over visual features from short video clips that are detached from the underlying physical space and capture only what is immediately visible. To facilitate human-centric environment understanding, we present an approach that links egocentric video and the environment by learning representations that are predictive of the camera-wearer's (potentially unseen) local surroundings. We train such models using videos from agents in simulated 3D environments where the environment is fully observable, and test them on human-captured real-world videos from unseen environments. On two human-centric video tasks, we show that models equipped with our environment-aware features consistently outperform their counterparts with traditional clip features. Moreover, despite being trained exclusively on simulated videos, our approach successfully handles real-world videos from HouseTours and Ego4D, and achieves state-of-the-art results on the Ego4D NLQ challenge. Project page: https://vision.cs.utexas.edu/projects/ego-env/

## 1 Introduction

Egocentric video offers a unique view into human activities through the eyes of a camera-wearer. Understanding this type of video is core to building augmented reality (AR) applications that can provide context-relevant assistance to humans based on their activity. Ego-video is thus the subject of several recent datasets and benchmarks that are driving new research [47, 75, 13, 26].

A key feature of the egocentric setting is the tight coupling of a camera-wearer and their persistent physical environment, i.e., a person's mental model of their surroundings informs their actions. This mental model is important, for example, to reach for a cabinet door out of view, to re-visit the couch to search for a misplaced phone or to visit spaces configured to support certain activities. This raises an important need for human-centric environment understanding — to learn representations from video that capture the camera-wearer's activities *in the context of their environment*. Such representations would encode the human-environment link, and allow models to jointly reason about both (e.g., to answer "what did the person cut near the sink?"). See Fig. 1.

Despite its importance, there has been only limited work on learning human-centric environment representations. Current video models segment a video into short clips (1-2s long) and then aggregate clip features over time (e.g., with recurrent, graph or transformer-based networks) for tasks like action forecasting [24, 26, 57, 25], temporal action localization [95, 50, 51, 96], episodic memory [26, 15], and movie understanding [90]. Critically, the clip features encode what is immediately visible in a short time window, and their aggregation over *time* does not equate to linking them in *physical space*. Other approaches use explicit camera pose information (e.g., from SLAM) to localize the camera-wearer, but not its relation to surrounding objects (e.g., forecasting [58, 68, 27]) or to group activities by location, but do not learn representations for agent video (e.g., affordance prediction [67, 57]).

37th Conference on Neural Information Processing Systems (NeurIPS 2023).

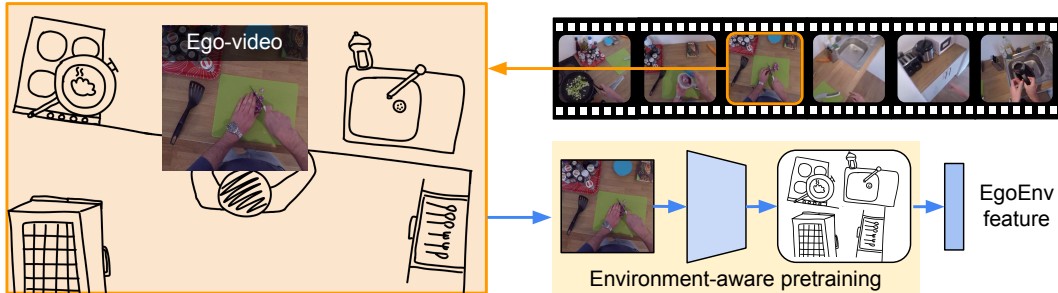

**Figure 1: Main idea.** Video models trained on short egocentric clips capture a narrow, instantaneous view of human activity (e.g., cutting an onion at the counter) without considering the broader context to which the activity is tied (e.g., the pan to the left to cook the onions, the fridge further behind to store leftovers). We propose to ground video in its underlying 3D environment by learning representations that are predictive of their surroundings, thereby enhancing standard clip-based models with complementary *environment* information.

To address these shortcomings, we propose to learn environment-aware video representations that encode the surrounding physical space. Specifically, we define the *local environment state* at each time-step of an egocentric video as the set of objects (and their rough distance) in front, to the left, right, and behind the camera-wearer. See Fig. 2. We use this state as supervision to train a transformer-based video encoder model that aggregates visual information across a video to build an *environment memory*, which can be queried to predict the local state at any point in the video.

The local state captures the rough layout of objects relative to the camera-wearer. It is important for understanding physical space — it provides a semantic signal to localize people (e.g., in the living room, from the arrangement of couches, lamps and tables) — as well as human behavior, since people move towards layouts that support activities (e.g., stove-top areas, dressers). Predicting the local state thus involves capturing the natural statistics of object layouts across different homes and then translating contexts across environments to reason about new ones. Once trained, given an observation from a new video, our model produces a drop-in EgoEnv feature which encodes *environment* information to complement the *action* information in existing video clip features.

An important practical question is how to supervise such a representation. Sourcing local state labels requires agent and object positions and omnidirectional visibility at each time step. This is challenging as egocentric videos only offer sparse coverage of the environment. Furthermore, they are prone to object detection, tracking, and SLAM failures. Therefore, for training we turn to videos generated by agents in simulation. This allows us to sample diverse, large-scale trajectories to cover the environment, while also providing ground-truth local state. Once trained with simulated video, we apply our models to *real-world* videos from new, unseen environments.

We demonstrate our EgoEnv approach on two video tasks where joint reasoning of both human action and the underlying physical space is required: (1) inferring the room category that the camera wearer is physically in as they move through their environment, and (2) localizing the answer to a natural language query in an egocentric video. These tasks support many potential applications, including AR systems that can offer context-relevant assistance.

We are the first to demonstrate the value of 3D simulation data for real-world ego-video understanding. Our experiments show that by capitalizing on both geometric and semantic cues in our proposed "local environment state" task, we can leverage video walkthroughs from *simulated agents* in HM3D scenes [64] to ultimately enable downstream human-centric environment models on *real-world* videos from HouseTours [7] and Ego4D [26]. Furthermore, models equipped with our EgoEnv features outperform both popular scene classification [97] and natural language video localization models [95], achieves state-of-the-art results on the Ego4D NLQ challenge leaderboard.

## 2 Related work

**Video understanding in 3D environments.** Prior work encodes short video clips [4, 85, 60, 33] or temporally aggregates them for additional context [89, 24, 95, 50, 51, 96, 90]. However, these methods treat the video as a temporal sequence and fail to capture the spatial context from the underlying persistent environment. For egocentric video, prior work has used structure from motion (SfM) to map people and objects for trajectory [58] and activity forecasting [27] and action grounding

in 3D [67, 14]. These approaches localize the camera-wearer but do not learn representations for the camera-wearer's surroundings. The model of [52] associates features to voxel maps to localize actions; however, they require a pre-computed 3D scan of the environment at both training and inference. Prior work groups clips by rough spatial location as topological graphs [57] or activity threads [61], but they stop short of learning representations using these groups. In contrast, we explicitly learn features that relate clips based on their spatial layout, for each step of an ego-video.

**Video representation learning.** Traditional video understanding methods learn representations by training models on large, manually curated video action recognition datasets [4, 54, 26]. Recent self-supervised learning (SSL) approaches eliminate the supervision requirement by leveraging implicit temporal signals [88, 55, 81, 40, 31, 84, 86, 87, 66, 23]. In contrast, we learn features that encode the local spatial-state of the environment (as opposed to temporal signals). Further, we show how to leverage state information that is readily accessible in simulation, but not in video datasets (i.e., locations and semantic classes of objects surrounding the agent) for training.

**Environment features for embodied AI.** In embodied AI, pose-estimates are used to build maps [11, 8, 62], as edge features in graphs [9, 7], as spatial embeddings for episodic memories [20], to project features to a grid map [28, 36, 5] or to learn environment features for visual navigation [65]. However, these approaches are explored solely in simulation and typically require accurate pose-estimates or smooth action spaces, and thus are not directly applicable to egocentric videos. Research on world-models [18, 29, 30] and unseen panorama reconstruction [41, 76, 44] *hallucinate* the effect of agent actions to aid decision-making in simulation. In contrast, we aim to learn environment features for an egocentric camera-wearer to aid real-world video understanding.

**Learning from simulated data.** Prior work has proposed cost-effective ways to generate large-scale synthetic image datasets for various vision tasks [77, 79, 16, 37, 19, 70]. In robotics, simulation environments have been developed to quickly and safely train policies, with the eventual goal of transferring them to real world applications [43, 80, 45, 6, 91, 72, 64, 78]. The resulting *sim-to-real* problem, where models must adapt to changes in simulator and real-world domains, is an active area of research for robot navigation [79, 42, 71, 3, 1]. However, simulated data for video understanding is much less explored. Prior work has synthesized data for human body pose estimation [12, 82, 93, 17], trajectory forecasting [48], and action recognition [69]. Rather than model human behavior, our approach is the first to directly capture the environment surrounding the camera-wearer for real-world video understanding tasks.

## 3 Approach

Our goal is to learn EgoEnv representations that encode the local surroundings of the camera-wearer. Such a representation would implicitly maintain a semantic memory of surrounding objects beyond what is immediately visible, and coupled with a standard video feature, would allow models to jointly reason about activities and the underlying physical space. Directly appending the camera pose with each video frame may capture the local state; however noise in pose estimated from ego-video with quick head motions and characteristic blur limits its utility (see Supp. for experiments).

Instead, we introduce an approach that leverages simulated environments where perfect state information is available to train models that can link visual information to the physical surroundings. To this end, we first define the local state prediction task in simulation (Sec. 3.1). Next, we introduce our transformer-based architecture that predicts the local state in videos (Sec. 3.2). Finally, we show how our model trained in simulation generates *environment features* for real-world egocentric video frames (Sec. 3.3).

### 3.1 Local environment state

We require a model that is aware of not just what is immediately visible in a single frame, but also of the camera-wearer's surroundings. We therefore define the *local environment state* of the camera-wearer as the set of objects in each relative direction — i.e., what objects are to the front, left, right or behind the camera-wearer, together with their rough distance from the camera-wearer — and train a model to predict this state. Our definition of local state takes inspiration from cognitive science [32, 46], and offers supervision signals that are both geometric (relative object locations) and semantic (semantic object labels), which we observe leads to strong representations.

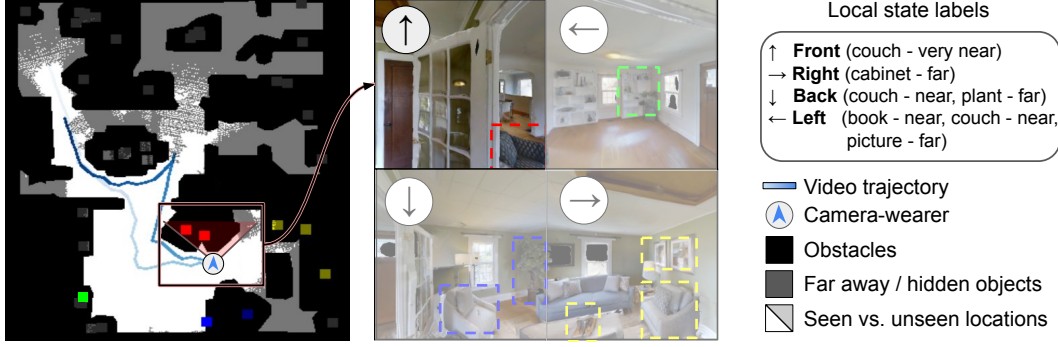

**Figure 2: Local environment state task. Left panel:** Top-down environment view showing the camera-wearer path (blue gradient line) and nearby objects (colored boxes). **Middle panel:** Egocentric view in each direction. Only the forward view (top left) is observed. Remaining views are shown for clarity. **Task:** Given an egocentric video of a person in their environment, the model must predict the set of objects (and their rough distance) in front, to the left, right, and behind the camera-wearer at each time-step (right panel). The model only sees the forward ego-view (middle panel, top left) and does not have access to the top-down map. Note that not all parts of the environment are seen during a walkthrough (white vs. grey regions on map) — models must link seen observations based on their shared space, as well as anticipate unseen surroundings based on statistics of training environments. Best viewed in color.

More formally, let $\mathcal{O}$ be a set of object classes. For a frame $f$ from a video trajectory in an environment, the local state is a tuple $(y_o, y_r)$. $y_o$ is a $4 \times |\mathcal{O}|$ dimensional matrix which represents instances of each object class in the four cardinal directions relative to the camera-wearer. $y_r$ is a matrix of the same size containing the distance of the objects in $y_o$ from the camera-wearer, discretized into 5 distance ranges between $0.25 - 5.0m$[1]. For direction $i$ and object class $j$, the labels are:

$$y_o[i,j] = \begin{cases} 1 & \text{if } d(p_a, p_j) < \delta \wedge \theta(p_a, p_j) = i \\ 0 & \text{otherwise,} \end{cases} \tag{1}$$

$$y_r[i,j] = \bar{d}(p_a, p_j) \quad \text{if } y_o[i,j] = 1, \tag{2}$$

where $p_a, p_j$ are the poses of the camera-wearer and object $o_j$ respectively, $d(p_a, p_j)$ is the euclidean distance between them, $\delta$ is a distance threshold for nearby objects (we set $\delta = 5.0$m, beyond which visible objects are small), $\theta(p_a, p_j) \in [0...3]$ is the discretized angle of the object relative to the agent's heading (forward, right, behind, left), and $\bar{d}(p_a, p_j) \in [0...4]$ is discretized distance. See Fig. 2 and Supp. for empirical analysis of related alternatives, e.g., predicting just object presence or image features.

We then train a model to predict the local state of the target video frame, conditioned on the video trajectory. Once trained, the model can relate what is visible in a frame to the camera-wearer's possibly *hidden* surroundings to produce environment-aware features.

Since supervision for camera-wearer pose and every object's location is non-trivial for egocentric videos, where camera localization and tracking is error prone, we leverage videos in simulated environments for training, as presented next.

### 3.2 Environment-aware pretraining in simulation

To source local state labels, we generate a dataset of video walkthroughs of agents in simulated 3D environments where agent and object poses are accessible at all times (see Sec. 4). We train our model in two stages described below.

#### 3.2.1 Pose embedding learning

While ground truth camera pose is available from the simulator at training time, a model trained to rely on it will fail on real-world egocentric video at test time, where pose estimates are noisy. With the goal of handling arbitrary indoor egocentric video, we instead explore representations that implicitly encode coarse pose information. See Supp. for pose-related experiments.

---

[1]For object classes with multiple instances, we select the nearest one.

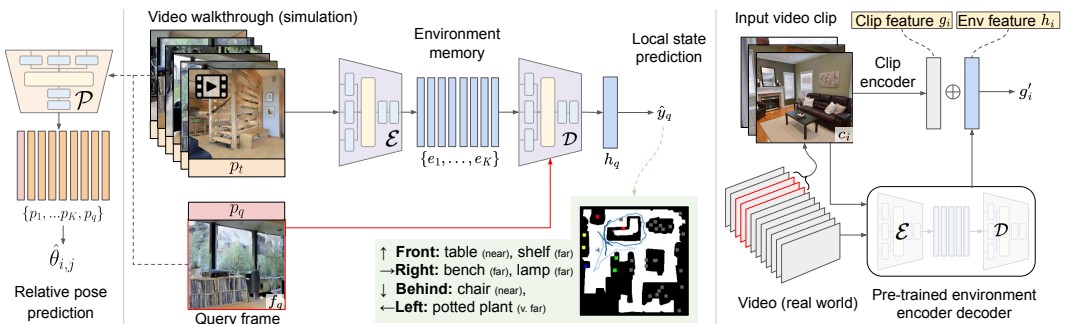

**Figure 3: Model framework. Left:** Our model first learns pose embeddings by predicting discretized relative pose between observations from simulated video walkthroughs (Sec. 3.2.1). **Center:** Next, it encodes observations and their pose embeddings into an *environment memory* that is trained to predict the local environment state for a query frame (Sec. 3.2.2). **Right:** Once trained, our model builds and queries the environment memory for any time-point of interest in a real-world video, to generate an environment feature for downstream video tasks in disjoint and novel scenes (Sec. 3.3). $\oplus$ = concatenation.

Specifically, for a sequence of RGB frames $\{f_t\}_{t=1}^T$ and camera poses $\{\theta_t\}_{t=1}^T$, we generate *pose embeddings* $\{p_t\}_{t=1}^T = \mathcal{P}(f_1, ..., f_T)$ using a transformer encoder network. These embeddings are used to predict the relative pose between each observation pair using a bilinear layer

$$\hat{\theta}_{i,j} = p_i^T V_p p_j + W_p^T(p_j - p_i) + b_p, \tag{3}$$

where $\hat{\theta}_{i,j}$ is the predicted relative pose and $V_p, W_p, b_p$ are trainable parameters. We discretize the relative pose into $12$ angles and $4$ distance ranges to provide an approximate yet robust pose estimate. The network is trained to minimize cross-entropy between the predicted and the target relative pose labels for all observation pairs $\sum_{i,j} \mathcal{L}_{ce}(\hat{\theta}_{i,j}, \theta_{i,j})$. The trained pose embeddings $p_t$ encode information to help relate video observations based on their location and orientation in the environment. Note that once trained, pose embeddings are inferred directly from video frame sequences — ground truth pose is only required for training.

### 3.2.2 Local state pretraining

Next, we train a model to embed visual information from a video walkthrough into an environment memory, which can then be queried to infer the local state corresponding to a given video frame from the same video. We implement this model as a transformer encoder-decoder model.

Specifically, for a video walkthrough $\mathcal{V}$ with RGB frames $\{f_t\}_{t=1}^T$, and a query frame $f_q$, we predict the local state $y_q = (y_o, y_r)$ as follows. First, pose embeddings $\{p_t\}_{t=1}^T$ are generated for the video and query frames following Sec. 3.2.1. Then, each frame is encoded jointly with the pose embedding using a linear transform $\mathcal{M}_p$.

$$x_t = \mathcal{M}_p([f_t; p_t]). \tag{4}$$

Next, we uniformly sample $K$ video frames to construct an environment memory using a transformer encoder $\mathcal{E}$, which updates frame representations using self-attention:

$$\{e_1, ..., e_K\} = \mathcal{E}(x_1, ..., x_K). \tag{5}$$

The resulting memory represents features for each time-step that contain propagated information from all other time-steps. Compared to prior work [24, 95, 50, 51, 96], our encoder has the ability to relate observations based on not just visual characteristics and their temporal ordering, but also their relative spatial layout in the environment. A transformer decoder $\mathcal{D}$ then attends over the memory using query $x_q$ to produce the output EgoEnv representation $h_q$:

$$h_q = \mathcal{D}(\{e_1, ..., e_K\}, x_q), \tag{6}$$

which is finally used to predict the local state using two linear classifiers $\mathcal{M}_o$ and $\mathcal{M}_r$ for object class and distance predictions respectively. The network is trained to minimize the combination of losses over the predicted and the target state labels for each direction:

$$\mathcal{L}(h_q, y_o, y_r) = \mathcal{L}_{bce}(\mathcal{M}_o(h_q), y_o) + \lambda \mathcal{L}_{ce}(\mathcal{M}_r(h_q), y_r),$$

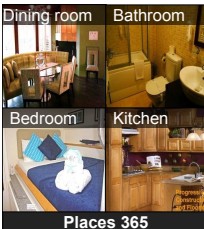 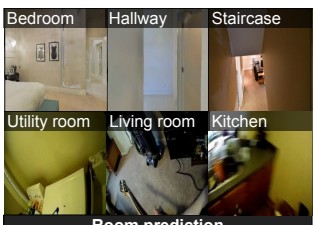 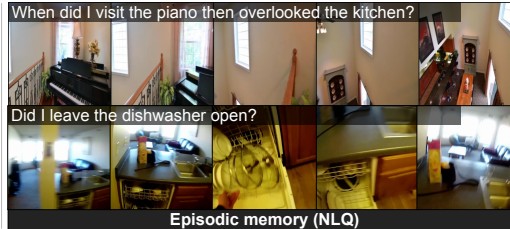

**Figure 4: Scene understanding in third-person photos vs. human-centric environment understanding.**
**Left:** Well-framed, canonical images from Places365 are considerably different from scene content observed in egocentric video. **Center and right:** Real egocentric video streams from HouseTours (top) and Ego4D (bottom) illustrating the value in modeling the underlying environment, rather than just what is visible in short clips. For example, the person does not explicitly look at the staircase while walking down it (center, top row); the spatial relation between the person, piano, and kitchen is important to answer the question (right, top row).

where $\mathcal{L}_{bce}$ and $\mathcal{L}_{ce}$ refer to binary cross-entropy and cross-entropy losses respectively. We set $\lambda = 0.1$ which balances the contributions of each loss function based on validation experiments. The distance loss is computed only for objects that are in the local state ($y_o = 1$). See Fig. 3 (left) and Supp. for more architecture details.

Learning to predict the local state involves aggregating information about observed objects across time, as well as anticipating unseen objects based on learned priors from the layout of objects in other scenes (e.g., TVs are usually in front of couches; kitchens have particular arrangements of sinks, refrigerators, and stove-tops).

Once trained, given a video in a new environment and a time-point of interest, our model constructs an environment memory, predicting the local state based on information aggregated throughout the test video. Importantly, $h_q$ — the EgoEnv feature — contains valuable information about the camera-wearer's surroundings, offering environment features to complement traditional video features (e.g., for a person watching TV, also encode the couch they are sitting on, the lamp nearby).

### 3.3 Environment-memory for video understanding with real videos

Next, we leverage our environment-memory model for real-world video tasks. A video understanding task defines a mapping from a sequence of video clips $\{c_1, ..., c_N\}$ from longer video $\mathcal{V}$ to a task label. We consider two tasks: (1) **ROOMPRED**: where the model must classify which room $r_t$ the camera-wearer is in (e.g., living room, kitchen) at time $t$ in the video, and (2) **NLQ**: natural language queries, an episodic memory task popularized recently in Ego4D [26] where the model must identify the temporal window $(t_s, t_e)$ in the video that answers an environment-centric query $q$ specified in natural language. See Fig. 4. Both tasks entail human-centric spatial reasoning from video.

Current models produce clip features that encode only what is immediately visible. This is sufficient for short-horizon tasks (e.g., action recognition), but as we will show, falls short on the tasks above that require extra reasoning about the agent's surroundings. Our environment-memory model addresses this by enhancing standard clip features with context from the camera-wearer's surroundings.

To do this, for each input clip in $c_i \in \mathcal{V}$, we select the center frame $f_i$ of the clip as the query frame. Following Sec. 3.2, we uniformly sample $K$ frames from the video around the query frame, encode them along with their pose embeddings (Eqn. 4), and build an environment memory using our environment encoder $\mathcal{E}$ (Eqn. 5). Finally, we use our decoder $\mathcal{D}$ to produce the output feature. This results in set of output EgoEnv features, one per input clip.

$$h_i = \mathcal{D}(\{e_1^i, ..., e_K^i\}, x_i). \tag{7}$$

Each environment feature then enhances the original clip feature as follows

$$g_i' = W_{\mathcal{E}}^T[g_i; h_i] + b_{\mathcal{E}}, \tag{8}$$

where $g_i$ is the original clip feature for clip $c_i$ (e.g., ResNet, SlowFast, EgoVLP) and $W_{\mathcal{E}}, b_{\mathcal{E}}$ are linear transform parameters. See Fig. 3 (right).

The new clip features $\{g_1', ..., g_N'\}$ consolidate features from what is directly visible in a short video clip and features of the (potentially unseen) space surrounding the camera-wearer. Put simply, our approach implicitly widens the field of view for tasks that reason about short video clips by providing a way to access features of their surroundings in a persistent, geometrically consistent manner.

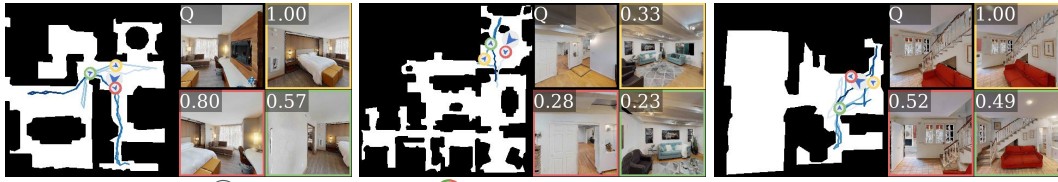

Video trajectory     Query frame pose     Top attended views

**Figure 5: Visualized attention weights.** Our model learns to attend to views that help solve the local state prediction task. The query frame $Q$ and top-3 attended views (colored boxes), their positions along the trajectory (colored circles), and their associated attention scores are shown. See Supp. for more examples.

Critically, we use the exact same EgoEnv representations to tackle both video understanding tasks. This is a departure from traditional sim-to-real approaches where a task-specific dataset needs to be carefully designed for every downstream task, which may be impractical. That said, given that today's 3D assets (Matterport3D, HM3D, etc.) focus on indoor spaces, our model is best suited to videos in indoor environments. We discuss the sim-to-real gap in detail in Supp.

## 4 Experiments

We evaluate how our EgoEnv features learned in simulation benefit real-world video understanding.

**Simulator environments** For training, we use the Habitat simulator [72] with photo-realistic HM3D [64] scenes to generate simulated video walkthroughs. We generate ~15k walkthroughs from 900 HM3D scenes, each 512 steps long, taken by a shortest-path agent that navigates to randomly sampled goal locations (move forward, turn right/left $30°$). For each time-step, we obtain the ground-truth local state from the simulator required in Sec. 3.1 (i.e., object labels and relative pose). For object labels, we map instance predictions across $|\mathcal{O}| = 23$ categories to the 3D scenes using a pretrained instance segmentation model [21] trained on the subset of scenes with semantic labels. Though the walkthroughs involve discrete actions, they share characteristics with real-world video (cameras at head-level; views covering the environment) making them suitable for transfer. See Supp. for examples and details.

**Video datasets** We evaluate our models on three egocentric video sources. (1) **HouseTours [7]** contains 119 hours of real-world video footage of house tours from YouTube. We use ~32 hours of video from 886 houses where the camera can be localized and create data splits based on houses. (2) **Ego4D [26]** contains 3k hours of real-world video of people performing daily activities. We use all videos annotated for the NLQ benchmark and apply the provided data splits, which yields 1,259 unseen scenes. (3) **Matterport3D (MP3D) [6]** contains simulated video walkthroughs from 90 photo-realistic 3D scenes. We use 146 long video walkthroughs and standard data splits [5].

These datasets provide an ideal test-bed for our approach. On the one hand, both HouseTours and Ego4D are *real-world* video datasets allowing us to test generalization to both real-world visuals as well as natural human activity across diverse, cluttered environments in unseen houses. On the other hand, MP3D offers novel scenes with distinct visual characteristics and object distributions compared to HM3D, allowing us to test our model's robustness to domain shift in a controlled *simulated* video setting. Note that none of the datasets have a 1-1 alignment in object taxonomy with HM3D, meaning our downstream tasks require generalization to both unseen environments and unseen objects.

We collect crowd-sourced labels for each task, which we will publicly share. For ROOMPRED these are room category labels from 21 classes (e.g., living room, kitchen) for each time-step on HouseTours and Ego4D. For NLQ these are natural language queries and corresponding response tracks in the video. On HouseTours, we crowd-source queries (e.g., "where did I last see my phone in the kitchen", "when did I first visit the bathtub") and on Ego4D, we use the official NLQ benchmark annotations, which require reasoning over actions, objects, and locations (e.g., "what tool did I pick up from the table", "where did I hang the pink cloth"). On MP3D, we source all labels directly from the simulator (9 room categories, and automatically generated NLQ queries from simulator object labels and locations). See Supp. for data collection details and Fig. 4 for examples.

**Experiment setup** For pre-training, we use 2048-D ImageNet-pretrained ResNet50 [35] features for each video frame. Our encoder-decoder models $\mathcal{P}, \mathcal{E}, \mathcal{D}$ are 2-layer transformers [83] with hidden dimension 128. $K = 64$ frames are sampled from each video to populate the memory. We train

models for 2.5k epochs and select the model with the lowest validation loss. For ROOMPRED, we generate a single EgoEnv feature aligned with the query clip. For NLQ we generate one feature per input clip. Full architecture and training details are in Supp.

**Baselines** For ROOMPRED we use a popular scene recognition model PLACESCNN [97] as the baseline model. For NLQ we use the state-of-the-art moment localization model VSLNET [94, 49, 53]. Within these two frameworks, we compare the following approaches to enhance clip representations: **FRAMEFEAT** uses a pretrained ResNet50 [35] model to generate a frame feature corresponding to each clip. **OBJFEAT** trains an object detector on all available simulated HM3D data and generates backbone features for each clip. We use the QueryInst model [21]. **MAE [34]** trains a state-of-the-art self-supervised learning approach to reconstruct masked patches of walkthrough video frames. **EGOTOPO [57]** trains a graph convolutional network (GCN) over the video graph built following [57]. **EPC [65]** trains an environment memory model to predict masked *zone* features conditioned on pose. **TRF (SCRATCH)** trains a scene-memory transformer model [20] that shares our model architecture but is randomly initialized and fine-tuned for the task.

These baselines represent various strategies to incorporate environment information into clip representations ranging from frame features (MAE, FRAMEFEAT, OBJFEAT), to topological graph-based features (EGOTOPO), to pose-based features (EPC). Note that OBJFEAT, MAE and EPC all pre-train on the same walkthrough videos as our approach. OBJFEAT further benefits from ground-truth object labels from the simulator. Features from these approaches augment the input clip representations following Eqn. 8 — baseline architectures remain unchanged. Note that EPC requires privileged information—ground-truth camera poses at inference time—whereas our model does not.

## 4.1 Pose embedding and local state pretraining

We begin by evaluating the pose embedding network trained to predict relative pose discretized into 12 angles and 4 distance ranges. On the validation set, the model achieves accuracies of 48.4% on relative distance prediction and 34.4% on relative orientation prediction. Note that this task is challenging — models must predict relative pose for all possible pairs of observations in a trajectory using their visual features alone — however the goal is to generate pose encodings, not to output perfect pose. Next, we evaluate how well our model can infer the local state, reporting average precision (AP) in each direction. Given a *forward* view, objects can be reliably recognized (37.8 AP) compared to naively outputting the distribution of objects seen at training (5.4 AP). Moreover, our approach can link views in the video trajectory to also infer and anticipate objects in other directions, i.e., to the right, left and behind (21.5, 24.9 and 20.2 AP respectively) given the forward view. We visualize the attention weights learned by our model to link relevant observations to the query in Fig. 5. Our model learns to select informative views for the task beyond just temporally adjacent frames or views with high visual overlap. For example, in the first image, the view with highest attention score (1.0) looks at the bed directly *to the left* (yellow box), allowing our model to benefit from information beyond its field of view.

## 4.2 EgoEnv features for room prediction

Next, we evaluate our method on predicting what room the camera-wearer visits in the video. All models have access to the full video, but inference is at at each time-step. The PLACESCNN model is a 2-layer MLP classifier trained on features from a Places365 [97] scene classification model. Features are max-pooled across a clip of $N = 8$ frames around the time-step of interest for additional context, as a single frame may be uninformative (e.g., facing a wall). We generate an environment feature aligned with the center of the clip.

Fig. 6 shows the results. We report top-1 accuracy as a function of dataset difficulty, measured by the prediction entropy of the Places365 model trained on canonical scene images. Instances are "hard" (high entropy) where frame-level information is insufficient for predicting the room type. Accuracy on the full dataset is at the far-right of the plots. All models perform better on HouseTours compared to MP3D and Ego4D since the house tours were captured explicitly to provide informative views of each room. Despite having access to all additional pre-training videos and labels as supervision, frame-level features from OBJFEAT and MAE prove to be insufficient for environment-level reasoning. All methods except ours perform worse with the introduction of hard instances where the surrounding environment-context is important (left to right). This is a key result: despite training our models entirely in simulation, and with videos from a set of disjoint environments, our EgoEnv features are useful for downstream tasks on real-world videos.

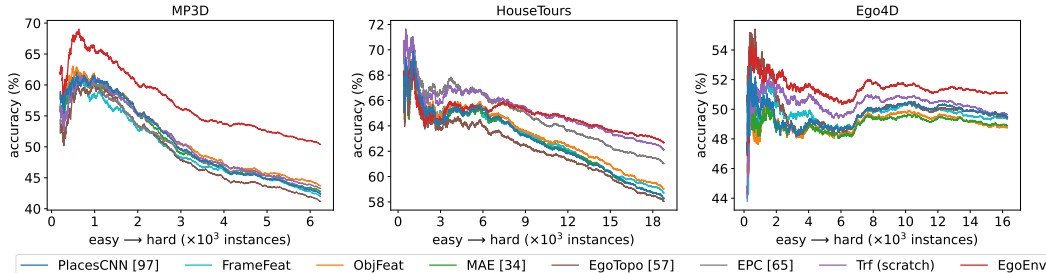

**Figure 6: ROOMPRED accuracy by instance difficulty.** Our method outperforms baselines, especially on *hard* instances (smaller performance drop along curve). EPC requires pose at inference, which is unavailable in Ego4D. Results are aggregated across three runs. See Supp. for averaged results with error bars.

| RANK1@M → | MP3D [6] | | | HouseTours [7] | | | Ego4D‡ [26] | | |
|---|---|---|---|---|---|---|---|---|---|
| | @0.3 | @0.5 | AVG | @0.3 | @0.5 | AVG | @0.3 | @0.5 | AVG |
| VSLNET [94] | 33.69 | 22.83 | 28.26 | 42.94 | 27.68 | 35.31 | 5.45 | 3.12 | 4.29 |
| FRAMEFEAT | 35.23 | 24.57 | 29.90 | 48.45 | 32.06 | 40.25 | 5.58 | 3.28 | 4.43 |
| OBJFEAT | 37.20 | 26.33 | 31.76 | 47.74 | 32.49 | 40.11 | 5.76 | 3.43 | 4.59 |
| MAE [34] | 35.11 | 24.13 | 29.62 | 44.49 | 27.82 | 36.16 | 5.65 | 3.02 | 4.34 |
| EGOTOPO [57] | 36.10 | 25.06 | 30.58 | 43.36 | 27.97 | 35.66 | 5.45 | 3.19 | 4.32 |
| EPC† [65] | 36.85 | **27.64** | 32.24 | 43.22 | 27.82 | 35.52 | - | - | - |
| TRF (SCRATCH) | 34.18 | 24.65 | 29.42 | 40.54 | 22.32 | 31.43 | 5.25 | 3.12 | 4.19 |
| EGOENV | **38.18** | 26.85 | **32.52** | **51.98** | **34.18** | **43.08** | **6.04** | **3.51** | **4.77** |

**Table 1: NLQ results.** All baselines build on VSLNet [95] with alternate features. †Privileged access to pose at inference, unavailable to our model, and absent in Ego4D. ‡Validation split. See Table 2 for test set results.

## 4.3  EgoEnv features for episodic memory

Next we evaluate on localizing the responses to natural language queries in egocentric video. We use VSLNET [94] and provide it with $N = 128$ clips sampled uniformly from the full video to generate predictions. We use SlowFast [22] clip features and generate environment features aligned with each input clip. We use the benchmark-provided metric of *Rank n@m*, which measures temporal localization accuracy [26].

Table 1 shows the NLQ results. Similar to ROOMPRED, instances are harder in MP3D than in HouseTours as MP3D's shortest-path agents produce moments that are quick transitions between objects and locations, and contain only short glimpses of them. The global, video-level information from EGOTOPO improves performance slightly on all datasets. Strong image-level supervision (object labels in OBJFEAT and FRAMEFEAT) results in the largest improvements; MAE, which has access to the same data but trains self-supervised representations, does not show strong improvements. EPC performs well on MP3D, where accurate pose is available from the simulator, but not on HouseTours with only noisy pose estimates (see Supp.). Our EgoEnv approach performs the best overall, outperforming even EPC, which (unlike EgoEnv) has access to ground-truth pose at inference.

Finally, Table 2 shows official eval server results for the Ego4D NLQ benchmark. We augment the baseline [63] with our EgoEnv features. We achieve state-of-the-art results, ranking 1st on the public leaderboard at the time of submission, and currently ranked 3rd. Note that Ego4D contains in-the-wild videos of natural human activity in diverse scenes (e.g., workshops, gardens) compared to the simulated walkthrough videos in pretraining. Due to this *sim-to-real* gap, our approach performs even better on instances aligned with training videos (navigation in indoor homes) as our Supp. experiments show. Our leading results on the full challenge set for this major benchmark demonstrates the value of our environment-centric feature learning approach, despite this gap.

## 4.4  Analysis of sim-to-real gap

Next, we discuss our approach in the context of the sim-to-real gap. Ego4D videos are in-the-wild, capture natural human actions and object-interactions, and take place in diverse scenes. These scenes may be significantly different from the simulated environments used for pre-training (navigation in indoor houses). In Table 3, we show results on the subset of videos that are aligned with the training environments on the Ego4D NLQ validation set. We select these using the scenario labels provided in Ego4D including indoor home scenarios (e.g., listening to music, household management)

| RANK 1@M → | @0.3 | @0.5 | AVG |
|---|---|---|---|
| CONE [39] | 15.26 | 9.24 | 12.25 |
| BADGERS@UW-MAD. [56] | 15.71 | 9.57 | 12.64 |
| INTERNVIDEO [10] | 16.46 | 10.06 | 13.26 |
| NAQ [63] | 21.70 | 13.64 | 17.67 |
| EGOENV | **23.28** | **14.36** | **18.82** |

Table 2: **Ego4D NLQ challenge results.** Our model obtains the best results against published approaches, and ranks 3rd among concurrent, unpublished work [38, 74].

and navigation-heavy scenarios (walking indoors and outdoors) while excluding outdoor activities (e.g., golfing, outdoor cooking). See Supp. for the full list of scenarios. Our approach shows healthier improvements across both sets of scenarios, highlighting the effect of the sim-to-real gap. However, despite this gap, our approach is still able to outperform other approaches over all scenarios, demonstrating the value of our environment-centric feature learning approach.

| RANK K@M → | R1@0.3 | R1@0.5 | AVG |
|---|---|---|---|
| NAQ (ALL) [63] | 24.12 | 15.04 | 19.58 |
| EGOENV (ALL) | **25.37** | **15.33** | **20.35** |
| NAQ (INDOOR) [63] | 28.91 | 17.97 | 23.44 |
| EGOENV (INDOOR) | **31.22** | **19.09** | **25.16** |
| NAQ (NAV) [63] | 23.58 | 15.49 | 19.53 |
| EGOENV (NAV) | **26.16** | **16.01** | **21.08** |

Table 3: **Ego4D NLQ validation set results on aligned scenes.** Our approach performs better on the subset of videos that are aligned with our approach's simulated training environments (navigation, indoor houses).

## 4.5 Ablation experiments

Next, we discuss some important ablations of our model design. We present full details of these experiments in Supp E and F, but we discuss the main conclusions here.

**Importance of pose information:** We measure the effect of pose embeddings on our local state prediction task. Our models show small improvements in predicting objects in the forward view (+0.7 mAP) where scene information is directly visible, but large for other views that need to be inferred: mAP improvements of +2.2 (right), +0.9 (behind) and +1.0 (left). Further, we directly embed ground-truth pose as part of the input and see benefits on both tasks on MP3D, but not on HouseTours, due to noise in extracted pose (compared to simulator-provided pose in MP3D).

**Alternate pretraining task formulations:** Next, we investigate alternate pretraining objectives instead of local state prediction including variants that only predict the object categories in each cardinal direction, but not the distances or that directly predicts the image features in each cardinal direction, among others. We find that our approach that requires predicting both object labels, orientations as well as rough distances offers a balance of both cues during pretraining, translating to strong downstream performance.

**Hyperparameter ablations:** We vary window size, memory size and the loss weighting term. We find that small window sizes are sufficient for localizing the room category for ROOMPRED, while larger windows are required for NLQ; and that the model is not very sensitive to the other parameters.

## 5 Conclusion

We proposed a framework to learn environment-aware representations in simulation and transfer them to video understanding tasks on challenging real-world datasets. Our approach outperforms state-of-the-art representations for predicting visited rooms and retrieving important moments from natural language queries, despite a significant sim-to-real gap. Despite its strengths, there are several opportunities for future work to improve our model further. These include incorporating 3D information into the local state task (currently defined in a 2D, top-down map), generating more *human-like* simulated videos and integrating more explicit approaches to tackle the sim-to-real gap.

**Acknowledgements** Thanks to Fu-Jen Chu and Jiabo Hu for help collecting the HouseTours annotations. UT Austin is supported in part by IFML NSF AI institute. KG is paid as a research scientist at Meta.

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
