# Supplementary material

This section contains supplementary material to support the main paper text. The contents include:

# A    Videos illustrating our approach

We include our supplementary video on our project page https://vision.cs.utexas.edu/projects/ego-env/. The video illustrates the local state prediction task (Sec. 3.2), downstream video tasks (Sec. 3.3) and our main results.

In the first part of the video, we demonstrate the local state prediction task from Sec. 3.2 of the main paper. The video shows the first-person view of the camera-wearer (left panel). The right panel shows the top-down view of the environment with the agent trajectory (blue gradient) and nearby objects (colored squares). Note that models only see the egocentric view — the top-down map is for illustration only. Given a simulated video walkthrough and a query time-stamp, the model must predict the direction and rough distance of each object near it. Correct, missing and false positive predictions are shown for each direction (left panel). Correct predictions on the top-down map are highlighted in cyan.

In the second part of the video, we show examples of the two downstream video tasks from Sec. 3.3 of the main paper, on MP3D, HouseTours and Ego4D. In the ROOMPRED examples, the model must predict which room the camera-wearer is in from a short clip. As mentioned in Sec. 4.2, the clips show quick motions and often contain ambiguous views making it hard to predict room labels directly using traditional scene recognition methods. For example, the "staircase" is not visible as the person descends it (Ego4D, bottom right video). In the NLQ examples, the model must predict the moment in time that answers a particular environment-centric query. The video examples show how this requires reasoning about the camera-wearer's surroundings. For example, in the HouseTours clip ("when did I visit the sink in the bathroom?") the sink is only seen briefly in the video but the response requires the window of time that the camera-wearer was physically near it (within arms reach) regardless of visibility. In the Ego4D clip ("Did I leave the drawer open"), the model must know where the drawer is relative to the camera-wearer and link their actions to this physical location in order to respond.

| Rank | Participant team | r@1, IoU=0.3 (↑) | r@1, IoU=0.5 (↑) | Mean r@1 (↑) | r@5, IoU=0.3 (↑) | r@5, IoU=0.5 (↑) | Last submission at | Meta Attributes |
|---|---|---|---|---|---|---|---|---|
| 1 | GroundNLQ | 25.67 | 18.18 | 21.93 | 42.06 | 29.80 | 5 months ago | View |
| 2 | asl_nlq | 24.13 | 15.46 | 19.79 | 34.37 | 23.18 | 5 months ago | View |
| 3 | ego-env | 23.28 | 14.36 | 18.82 | 27.25 | 17.58 | 5 months ago | View |
| 4 | mzs (SnAG) | 21.90 | 15.43 | 18.67 | 38.29 | 27.05 | 5 months ago | View |
| 5 | VioletPanda (finetuned+vgit+git) | 22.03 | 14.11 | 18.07 | 26.50 | 17.81 | 5 months ago | View |
| 6 | Host_90322_Team (NaQ++ ReLER) | 21.70 | 13.64 | 17.67 | 25.12 | 16.33 | 7 months ago | View |

**Figure 1: Snapshot of the Ego4D NLQ leaderboard.** Leaderboard can be found on the challenge page. Note that both GroundNLQ [38] and asl_nlq [74] are concurrent works that were added to the leaderboard after the submission of this paper.

## B  Sim-to-real gap in Ego4D

As mentioned in Sec. 4.4 of the main paper, our model is affected by the type and diversity of pretraining data — videos of simulated agents walking around a house — limiting its generalization to unconstrained real-world video. Similarly, our approach is limited by simulator functionality — HM3D scenes support a small set of objects, which may not overlap with real-world environments, and Habitat does not support fine-grained object-interactions (e.g., chopping vegetables). As a result, we find that our approach works well on videos that are consistent with pretraining (i.e., indoor home scenarios; videos with lots of walking and less object interaction) but contributes less on out-of-distribution scenes and activities (e.g., golfing, outdoor cooking). The full list of scenarios is in Table 1.

Our results on the benchmark challenge (test set) in Table 2 corroborate this result. Note that our method was the top-ranked approach at the time of submission. Since then, other unpublished methods have been submitted to the leaderboard. We expect that future advancements in simulator capabilities (e.g., human motion models for agents, fine grained object interaction simulation) will help address this class of limitations. Fig. 1 shows a snapshot of the leaderboard as of 10/23/23.

| Indoor | Visiting exhibition, On a screen (phone/laptop), Listening to music, Household management - caring for kids, Talking on the phone, Watching tv, Talking to colleagues, Electronics (hobbyist circuitry board kind, not electrical repair), Practicing a musical instrument, Eating, Cooking, Making coffee, Playing board games, Working at desk, Working out at home, Reading books, Playing games / video games, Hosting a party |
|---|---|
| Navigation | Roller skating, Bus, Walking on street, Indoor Navigation (walking), Walking the dog / pet, Car - commuting, road trip, Grocery shopping |

**Table 1: List of scenarios aligned with training environments.**

## C  Data collection and annotation details.

We present data collection and annotation details for simulated walkthroughs, and our two downstream tasks (RoomPred and NLQ) for all three datasets (MP3D, HouseTours, Ego4D).

### C.1  Walkthrough generation details

As mentioned in Sec. 4 (simulators) of the main paper, we generate simulated walkthroughs in HM3D [64] scenes to train our models. Given an environment, we first cluster all navigable points using KMeans, selecting between 4-64 clusters depending on the environment size. With each cluster

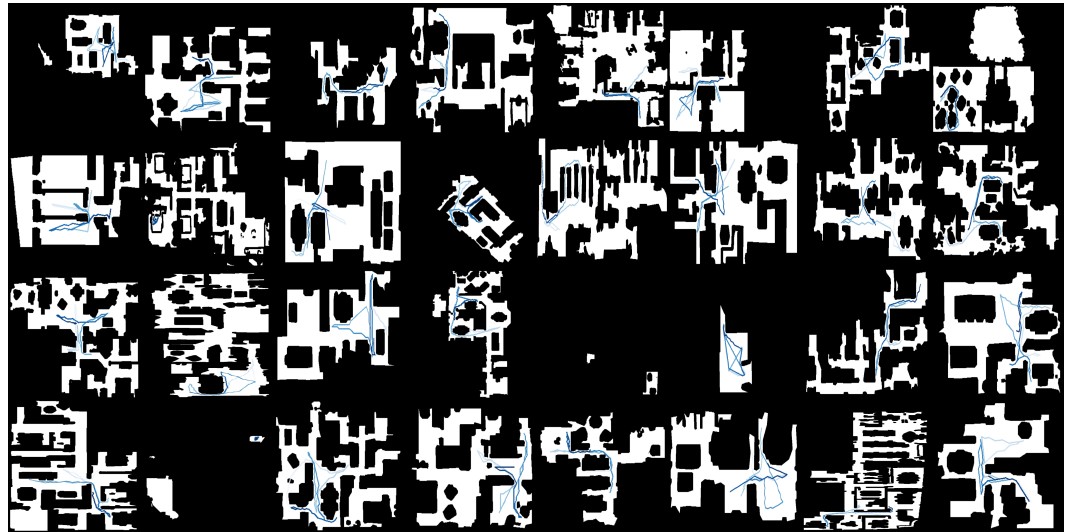

**Figure 2: Walkthrough examples in HM3D.** The blue gradient represents the trajectory from start (white) to end (blue). Black and white regions represent obstacles and free space respectively.

|  | SCENES | ROOMPRED | NLQ |
|---|---|---|---|
| HM3D [64] | 800 / 100 / – | – | – |
| Matterport3D [6] | 57 / 6 / 21 | 1337 / 173 / 536 | 8380 / 837 / 3452 |
| HouseTours [7] | 570 / 135 / 181 | 4512 / 1107 / 1593 | 2009 / 462 / 706 |
| Ego4D [26] | 998 / 328 / 333 | 3604 / 1596 / 1434 | 11291 / 3874 /4004 |

**Table 2: Dataset train / val / test splits.** Splits based on scenes and instances per downstream task are shown. HM3D is used only for pretraining (Sec. 3.2).

centroid as a starting location, we sample 8-16 nearest goal locations, shuffle them (to allow revisitation), and make an agent visit the goals in sequence. We use a shortest-path planning agent that uses the underlying navigation graph to reach goals in the fewest number of steps. We collect a dataset of ~15k episodes, each of 512 steps, of our agents visiting such goal sequences for experiments in Sec. 3.2. Fig. 2 shows a random sample of walkthroughs.

Note that the walkthroughs are generated in environments where objects are not moved, however a large part of real-world environments are in fact static. This includes static scene elements like doors, windows, counter tops, staircases, and most objects that are typically not moved like refrigerators, beds, couches, TV sets. Encoding these objects and scene elements can thus still provide value for human-centric environment understanding, even when some objects may have moved around.

## C.2 Data collection and annotation details

As mentioned in Sec. 4 (simulators) of the main paper, we collect labels for each video dataset. For Matterport3D, we directly use the ground truth information available through the simulator to extract labels. For HouseTours and Ego4D, we crowd-source annotations where required. We describe the data collection process and present data statistics for each dataset and task. The resulting dataset splits can be seen in Table 2.

### C.2.1 Annotation requirements

**ROOMPRED** For this task, room labels are required at each time-step of the video. The 9 room categories used in Matterport3D are in Table 3 (left). These categories are pre-defined in the simulator. The 21 room categories used in HouseTours and Ego4D are in Table 3 (right). These categories were generated manually from a combination of Matterport3D room categories and a relevant subset of Places365 categories corresponding to indoor scenes.

**NLQ** For this task, natural language queries and corresponding moment boundaries (start and end times) are required. For HouseTours, we define 7 query templates where $o$ refers to objects and $r$

Table 3 (Matterport3D):

| hallway | bathroom | bedroom |
|---|---|---|
| office | kitchen | living room |
| lounge | dining room | family room |

Table 3 (HouseTours / Ego4D):

| attic | balcony | basement | bathroom | bedroom |
|---|---|---|---|---|
| closet | corridor/hallway | dining room | driveway | front door/entrance |
| garage/shed | gym | kitchen | lawn/yard/garden | living room |
| office/home office | porch | recreation room (billiards room/play room) | | |
| staircase | storage/laundry/utility room | swimming pool | | |

**Table 3: Room taxonomies for Matterport3D, HouseTours and Ego4D**

| Template | Example | Description |
|---|---|---|
| see $o$ | Where did I first see the remote control? | Objects must be visible for the moment duration |
| see $o$ in $r$ | When did I see the mirror in the bathroom? | Object must be physically inside the room |
| see $o_1$ then $o_2$ | Where did I see a table then a chair? | Objects can either be seen together or in quick succession |
| visit $r_1$ then $r_2$ | When did I walk from the living room to the kitchen? | Start = when the person begins to leave; End = they are fully inside the kitchen. |
| visit $o/r$ | When did I last visit the couch?; When did I last visit the bedroom? | Visit = physically near an object (within arms reach) or physically inside a room |
| visit $o_1$ then $o_2$ | When did I visit the lamp then the couch? | Same as see $o_1$ then $o_2$, but using the "visit" criteria above |
| visit $o$ in $r$ | When did I visit the mirror in the bathroom? | Same as see $o_1$ in $o_2$, but using the "visit" criteria above |

**Table 4: Query templates and examples for the NLQ task on MP3D and HouseTours.**

refers to rooms: "see $o$", "see $o$ in $r$", "see $o_1$ then $o_2$", "visit $r_1$ then $r_2$", "visit $o/r$", "visit $o_1$ then $o_2$", "visit $o$ in $r$". Each template captures a type of question that requires a different mechanism of reasoning. "see" queries require reasoning about what is immediately visible; "visit" queries require an understanding of where the camera-wearer is in the environment and what objects are nearby (within arms reach); "see/visit $o_1$ then $o_2$" and "see/visit $o$ in $r$" require both spatial and temporal reasoning. Natural language queries follow from these templates. For example, for the "visit $r_1$ then $r_2$" template, a natural language query may be "When did I first walk from the kitchen to the bathroom?". The list of query templates, examples and descriptions can be seen in Table 4. The task definition follows prior work [26] but is adapted for the datasets used, and contains more environment-centric queries. For Ego4D, we use the existing NLQ benchmark task annotations.

Our video in Sec. A shows examples of both tasks to complement the image examples from Fig. 4 of the main paper. The video highlights the stark contrast between prediction in static images (third-person photos) which contains well-framed images that are easy to recognize, and egocentric video which is much more challenging. In this setting, video is tied to quick ego-motion as the camera-wearer moves around the environment and objects are seen only briefly (or not at all) in non-canonical viewpoints.

### C.2.2   Matterport3D annotations

For ROOMPRED, navigable location are mapped to room categories using information from the simulator. We map agent positions for each video frame to these categories. For NLQ we use room labels and extracted object positions to generate queries from the 7 templates above. We define objects as "seen" if they occupy at least 5% of the pixels in a given frame. We define objects as "visited" if the agent is $< 1.0$m from the object, regardless of its visibility, following embodied navigation protocol (ObjectNav [2]). Rooms are "visited" using the position $\rightarrow$ room category mapping above. We generate queries for each template by tracking objects and rooms that are seen and visited over time. If there are multiple visitations to an object or room, we ensure unique responses to queries by adapting them to consider only the first (or last) visit.

Fig. 3 shows annotation statistics for both tasks on Matterport3D. Trajectories are fixed length (512 steps) with 7 room transitions on average. Both visits and moments are short, making it difficult to localize the response to the natural language query, and providing little extra context to recognize rooms. See our video in Sec. A for examples.

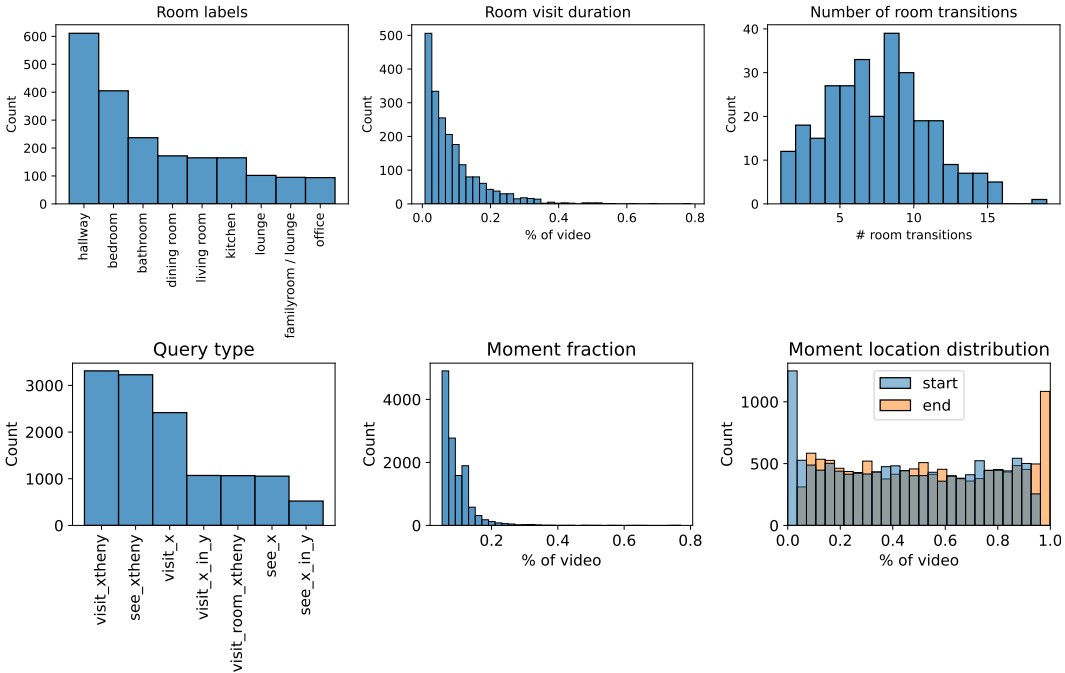

**Figure 3: Annotation statistics for Matterport3D. Top panel:** ROOMPRED data distribution. (left) distribution of room categories; (center) length of each room visit relative to the full video; (right) number of room transitions in each video. **Bottom panel:** EPISODIC MEMORY RETRIEVAL data distribution. (left) distribution of query types; (center) length of each annotated moment relative to the full video; (right) distribution of start and end times of annotated moments.

### C.2.3 HouseTours annotations

We crowd-source annotations for real-world videos from HouseTours. For ROOMPRED, we ask annotators to watch a video and mark the start and end time of each "visit" to a room. They must then label each visit with one of the 21 room categories (from Table 3, right). An illustration of the annotation interface is shown in Fig. 6.

For NLQ, we ask annotators to identify an interesting moment (e.g., where a person sees a salient object, moves from one room to another, visits an important object) which serves as the answer to a query. The moment is specified by a start and end time, while the query is specified as natural language text generated following one of the 7 template classes from Table 4. An illustration of the annotation interface is shown in Fig. 7.

Fig. 4 shows annotation statistics for both tasks on HouseTours. In general, trajectories are relatively shorter than Matterport3D, though they involve a similar number of room transitions (6 on average). They share similar challenges with short moments. See our video in Sec. A for examples.

### C.2.4 Ego4D annotations

Following the same procedure as HouseTours, we crowd-source room visit labels on Ego4D videos (see Fig. 5). Ego4D videos are much longer (30 mins on average). As a consequence, visits are naturally a much smaller fraction of the overall video and the number of room transitions are much higher (16 on average). The distribution of room categories is similar to HouseTours. Annotation statistics for NLQ can be found in Section F.4 of the supplementary material of the Ego4D paper [26].

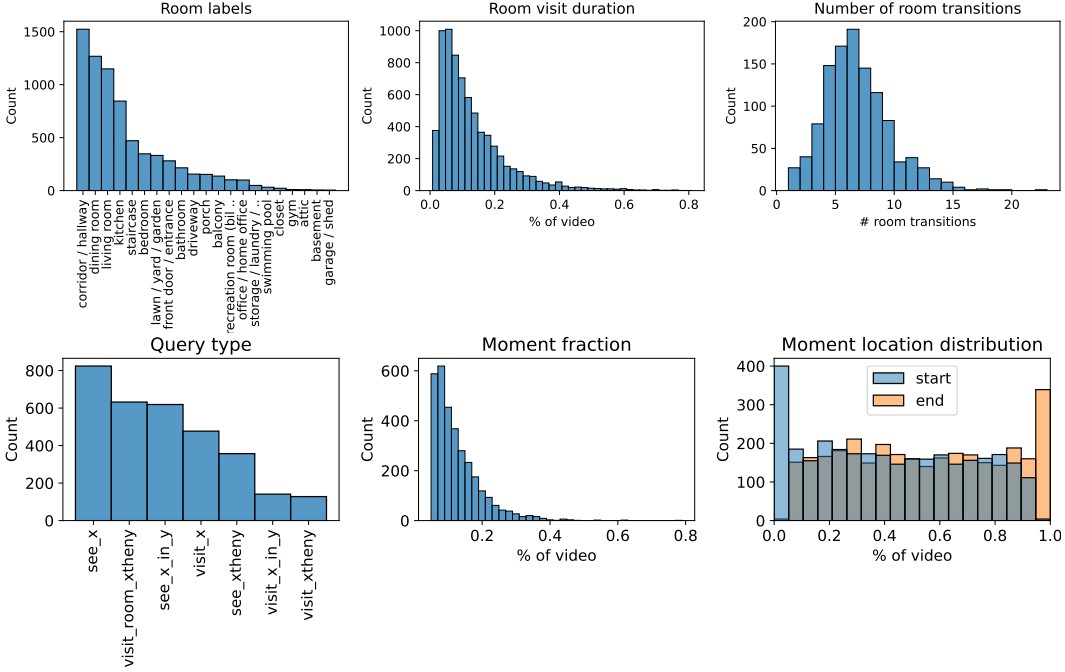

**Figure 4: Annotation statistics for HouseTours. Top panel:** ROOMPRED data distribution. (left) distribution of room categories; (center) length of each room visit relative to the full video; (right) number of room transitions in each video. **Bottom panel:** EPISODIC MEMORY RETRIEVAL data distribution. (left) distribution of query types; (center) length of each annotated moment relative to the full video; (right) distribution of start and end times of annotated moments.

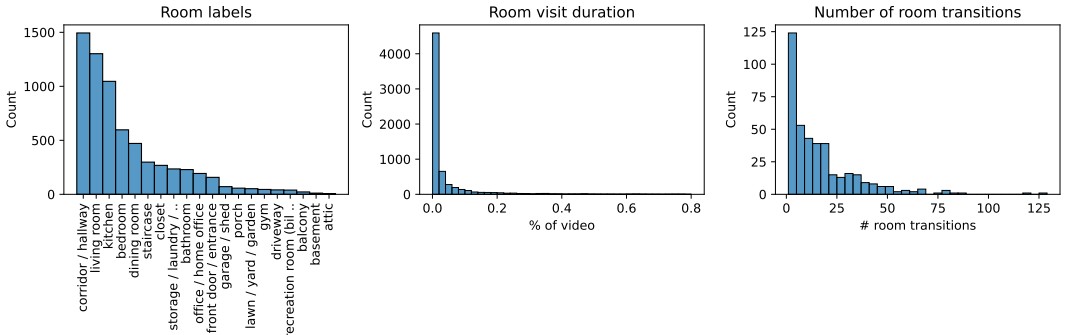

**Figure 5: Annotation statistics for Ego4D.** ROOMPRED data distribution. (left) distribution of room categories; (center) length of each room visit relative to the full video; (right) number of room transitions in each video. We do not collect annotations for NLQ on Ego4D. We use annotations from the official Ego4D benchmark challenge (Section F.4. in the Ego4D paper [26]).

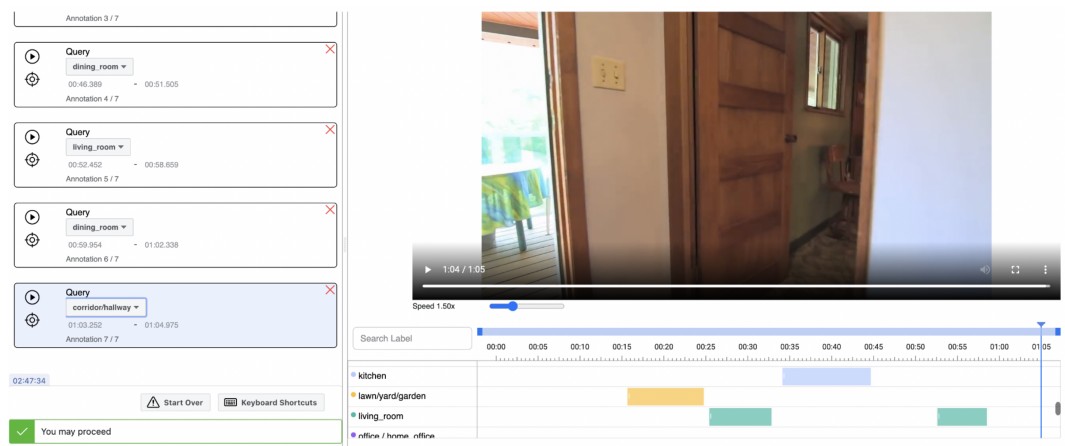

**Figure 6: Annotation interface for collecting ROOMPRED labels.** Annotators must densely segment room visits (start and end times) and associate a class label to each of them.

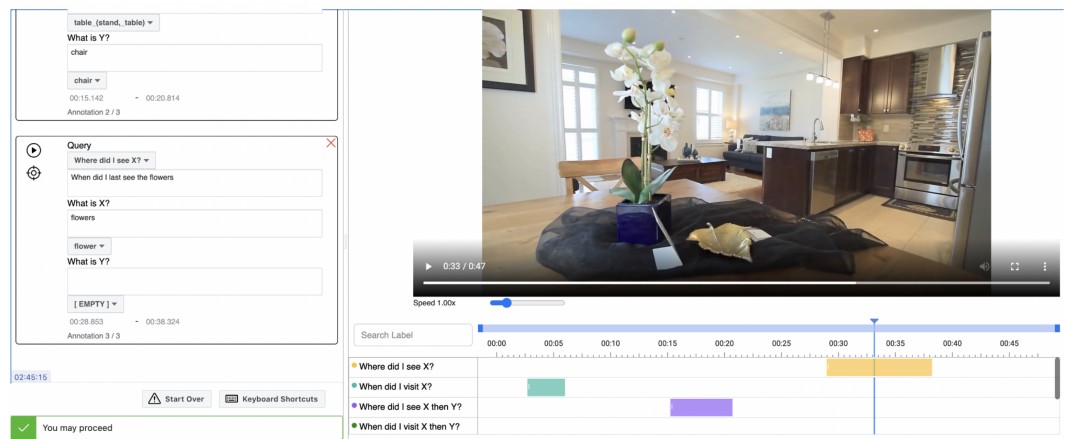

**Figure 7: Annotation interface for collecting NLQ labels.** Annotators must identify an interesting moment in time (start and end time) and associate a query template, a natural language query and object/room labels that fill the query template slots.

| chair | table | picture | cabinet |
|---|---|---|---|
| cushion | sofa | bed | chest_of_drawers |
| plant | sink | toilet | stool |
| towel | tv_monitor | shower | bathtub |
| counter | fireplace | shelving | seating |
| furniture | appliances | clothes | |

**Table 5: HM3D object taxonomy**

# D  Implementation and training details

We present architecture and training details for all approaches.

## D.1  Architecture and training details (pretraining)

We present additional details for our model in Sec. 3 of the main paper, as well as the baselines in Sec. 4 (baselines). The full list of hyperparameters are in Table 6.

**Pose embedder $\mathcal{P}$, environment encoder $\mathcal{E}$ and decoder $\mathcal{D}$.** We build on the transformer [83] architecture for our model. The inputs $f \in \mathbb{R}^{2048}$ are features from an ImageNet-pretrained ResNet-50 . These are encoded into 128-dimension vectors using a visual encoder (2-layer MLP).

First, 128-$D$ pose embeddings are generated following Eqn. 3. These pose embeddings are concatenated with the visual embedding, and then transformed back to a 128-dimension vector using $\mathcal{M}_p$. Sin-cosine position embeddings are added to this input following [83] resulting in the transformer input $\{x_1, ..., x_N\}$ in Eqn. 4. Note that a separate visual encoder generates input features for the encoder $\mathcal{E}$ and the pose embedder $\mathcal{P}$.

The encoder $\mathcal{E}$ performs multi-headed self-attention using these inputs, with 2 layers, 8 heads and hidden dimension 128. The decoder $\mathcal{D}$ is a 2-layer transformer with hidden dimension 128 which attends to the outputs of $\mathcal{E}$ to generate the output representation. We predict relative directions for 23 object classes in HM3D (see Table 5). We use the transformer implementation from PyTorch [59].

Our architecture is similar to prior embodied navigation approaches [65, 20] but does not require pose (and instead uses pose embeddings), includes the environment decoder that queries the memory based on pose embeddings and visual content, and is trained using our proposed learning objective in Sec. 3.2. See Table 6 (left).

**EPC architecture details.** EPC [65] is a transformer encoder-decoder model that masks out frames from physical locations and predicts the features of these masked zones given a query pose. We generate graphs to train this baseline following their approach.

Specifically, for every video frame $\{f_1, ..., f_T\}$, we compute the geometric viewpoint overlap with every other frame. The viewpoint overlap $\psi(f_i, f_j)$ is calculated by projecting pixels from the frames to 3D point-clouds using camera intrinsics, agent pose and depth measurements, and measuring the percentage of shared points across frames. We use $\psi(f_i, f_j)$ as our distance metric to cluster all frames in video into zones using hierarchical agglomerative clustering. We set the distance threshold to $0.8$ as larger values result in too few zones.

We sample 4 unseen zones per instance in a batch of which one is "positive" and three are "negatives" for contrastive learning. We collect negatives from all instances in a batch during training. The network is trained using noise-contrastive estimation following [65]. See Table 6 (center).

**EgoTopo architecture details.** EgoTopo [57] is an approach to translate egocentric video frames into a topological graph, where each node contains a list of clips that correspond to a physical location, and edges correspond to rough spatial layout.

For MP3D and HouseTours, ee directly use available pose information to determine whether two frames belong to the same zone or not (i.e., we do not train a retrieval network to approximate this). This amounts to a clustering of the trajectory based on pose (position and heading) and represents an enhanced version of EgoTopo that benefits from pose data. We use affinity propagation to cluster frames into nodes. To compute edges, we calculate the distance between the centroid of each node and

| Transformer architecture | |
| --- | --- |
| Input dim | 2048 |
| Hidden size | 128 |
| Pose emb dim | 128 |
| Walkthrough length | 512 |
| Memory size (K) | 64 |
| # attention heads | 8 |
| # encoder layers | 2 |
| # decoder layers | 2 |

| EPC parameters | |
| --- | --- |
| # Unseen zones | 4 |
| Clustering threshold | 0.8 |

| EgoTopo parameters | |
| --- | --- |
| Affinity prop damping | 0.5-0.9 |
| GCN hidden dim | 128 |
| # GCN layers | 2 |

| Optimization parameters | |
| --- | --- |
| Max epochs | 2500 |
| Learning rate | 1e-4 |
| Weight decay | 2e-5 |
| Batch size | 512 |

**Table 6: Architecture and training hyperparameters for pretraining (Sec. 3.2)**

assign an edge if this distance is $< 3.0$m to be consistent with Eqn. 2. For Ego4D, pose information is not available. We fall back to clustering visual frames based on ImageNet pre-trained ResNet-50 frame features.

Node features are calculated as the average of features assigned to that node. We use a 2-layer graph convolutional neural network (GCN) to aggregate features across nodes, and then average them to form a single video encoding following [57]. See Table 6 (center).

**MAE architecture details.** We use all frames from our generated walkthroughs to train an MAE VIT-large model with $16 \times 16$ patch size. We use the authors existing code and train a model for 200 epochs.

**ObjFeat architecture details.** We train a QueryInst instance segmentation model [21] with a R-50-FPN backbone, using a dataset sampled from the 100 HM3D scenes with ground-truth semantic object annotations. We use the implementation in the MMDetection package.

**Training details.** As mentioned in Sec. 4 (experiment setup) of the main paper, we train our models for 2500 epochs using the Adam optimizer with learning rate $1e-4$. We sample $K = 64$ frames to construct our memory. At training, we sample frames from the video but randomly offset frame indices to train robust models. During inference, we uniformly sample frames. We select the model with the lowest validation loss to evaluate downstream. See Table 6 (right) for all optimization hyperparameters.

## D.2 Architecture and training details (downstream)

We present additional details for the approaches in Sec. 3.3 of the main paper. The full list of hyperparameters are in Table 7.

**Room prediction models.** As mentioned in Sec. 4.2 of the main paper, for our PLACESCNN baseline model, we build a classifier on top of features from the wide ResNet-18 model from [97]. We use the authors existing code and their provided pretrained models to initialize the model. The classifier head is a 2-layer MLP with hidden dimension $512$. The backbone is frozen and only the classifier is fine-tuned. $N = 8$ frames around the target frame are used to provide additional context. The features are max-pooled before classification.

For our models, we use the target frame as the query to produce a single environment-feature, which is then concatenated with all $N = 8$ frames and aggregated following Eqn. 8. This new, enhanced input is fed into the baseline model as described above.

We train all models for 80 epochs with learning rate $1e-4$ using the Adam optimizer. The full list of hyperparameters are in Table 7 (left)

**Episodic memory retrieval models.** As mentioned in Sec. 4.3 of the main paper, we build on the VSLNet model from [94]. We use an existing implementation based on the authors original code. Visual inputs are encoded as $N = 128$ clip features, created by adaptive average pooling of SlowFast-R50 clip features. Note that for MP3D we use ResNet-50 features as the walkthroughs contain discrete agent steps, rather than smooth video frames.

| ROOMPRED | |
| --- | --- |
| # input frames | 8 |
| Hidden size | 512 |
| # layers | 2 |
| **Optimization parameters** | |
| Max epochs | 80 |
| Learning rate | 1e-4 |
| Weight decay | 2e-5 |
| Batch size | 512 |

| NLQ | |
| --- | --- |
| # Input clips | 128 |
| Hidden size | 128 |
| Highlight lambda | 5.0 |
| Extend boundary % | 0.1 |
| # heads | 8 |
| # layers | 4 |
| Dropout rate | 0.2 |
| **Optimization parameters** | |
| Max epochs | 200 |
| Learning rate | 1e-4 |
| Weight decay | 1e-2 |
| Batch size | 64 |

**Table 7: Architecture and training hyperparameters for ROOMPRED and NLQ (Sec. 3.3)**

For our models, we select the center frame of each of the $N = 128$ inputs and use them as query frames to produce $N = 128$ environment features. Each pair of input feature and environment feature is aggregated following Eqn. 8, and then input to the VSLNet model described above. Note that we perform this aggregation *after* the video affine and feature encoding layers in VSLNet.

We train all models for 200 epochs with a learning rate of $1e - 4$ using the Adam optimizer. The full list of hyperparameters are in Table 7 (right).

For Ego4D experiments, we use the hyperparameters described in the respective benchmark whitepapers [26, 49, 53] and only aggregate precomputed environment features as described above.

## E   Supplementary experiments and analysis.

### E.1   Experiments with extra pretraining data (Ego4D videos vs. simulator walkthroughs)

In Sec. 4 (baselines), all approaches have access to the same set of simulator-generated walkthrough videos for pre-training for apples-to-apples comparisons. In this section, we test the effect of directly performing self-supervised pretraining on Ego4D videos instead. In Table 8 we compare our MAE baseline trained on walkthrough videos with the same architecture trained on Ego4D frames [92] using the author provided pretrained model.

| RANK1@M → | @0.3 | @0.5 | AVG |
| --- | --- | --- | --- |
| MAE (WALKTHROUGH) | 5.65 | 3.02 | 4.34 |
| MAE (EGO4D) | 5.65 | 3.27 | 4.47 |
| EGOENV | **6.04** | **3.51** | **4.77** |

**Table 8: NLQ results for Ego4D with MAE trained on Ego4D.**

### E.2   Additional experiments with pose embeddings and ground-truth pose

As noted in Sec. 3.2.1 ground truth pose may be utilized directly by our model, however it is not easily available in real-world egocentric video, which makes our use of inferred pose embeddings a strength.

### E.2.1   Importance of pose embeddings

As noted in Sec. 3.2.1 of the main paper, we train pose embeddings to help relate observations by their visual content as well as relative orientation of capture. We measure the effect of pose embeddings on our local state prediction task. Our models see improvements in predicting objects in each of the cardinal directions (26.1 vs. 24.9 mAP) as well as improved object distance prediction (59% vs 58%). This improvement is small for predicting objects in the forward view (+0.7 mAP) where scene

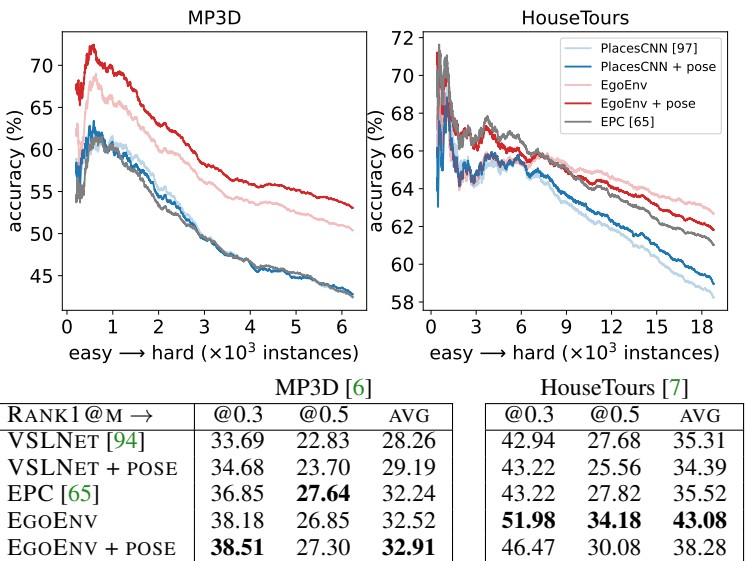

**Figure 8: Task performance using ground-truth pose.** ROOMPRED (top) and NLQ (bottom). Our approach (w/ and w/o pose) outperforms baselines. Noisy pose in HouseTours degrades performance. Ego4D videos do not have associated pose.

| RANK1@M → | MP3D [6] | | | HouseTours [7] | | |
|---|---|---|---|---|---|---|
| | @0.3 | @0.5 | AVG | @0.3 | @0.5 | AVG |
| VSLNET [94] | 33.69 | 22.83 | 28.26 | 42.94 | 27.68 | 35.31 |
| VSLNET + POSE | 34.68 | 23.70 | 29.19 | 43.22 | 25.56 | 34.39 |
| EPC [65] | 36.85 | **27.64** | 32.24 | 43.22 | 27.82 | 35.52 |
| EGOENV | 38.18 | 26.85 | 32.52 | **51.98** | **34.18** | **43.08** |
| EGOENV + POSE | **38.51** | 27.30 | **32.91** | 46.47 | 30.08 | 38.28 |

information is directly visible, but large for other views that need to be inferred: mAP improvements of +2.2 (right), +0.9 (behind) and +1.0 (left).

### E.2.2   Importance of ground truth pose

In this section, we explore using pose estimates obtained directly from the simulator or using off-the-shelf structure for motion methods. Note that the existing method EPC in Fig. 6 and Table 1 already uses this ground-truth pose information, which other models (including ours) do not have access to.

**Extracting pose information.** For HouseTours videos, we run COLMAP [73], a structure from motion framework to compute camera-pose information associated with each frame of the video. For this, we first extract frames from each video at 2fps. Then we run COLMAP using a precomputed vocabulary tree file (flickr100K_words32K). The resulting trajectories are inherently noisy due to the approximate nature of the SfM pipeline and the absence of true camera parameters for such in-the-wild video. We post-process them by removing erroneous pose values (ones that cause jumps in pose atypical to smooth motion).

Overall, COLMAP successfully localizes ~32 hours of video from 886 houses out of the original 119 hours available in [7]. Some examples of video trajectories are shown in Fig. 9. Comparing these to the simulated trajectories in Fig. 2, we see smoother trajectories overall, but with unrealistic jumps in localizations and loop-closure failures. Note that we visualize only the trajectory, not obstacles in the environment, as we do not have access to occupancy maps for the real-world videos.

Note that these videos are tours of indoor spaces with the intention of visually covering a large area. As a result, the camera-wearer moves slowly and smoothly to show parts of the house. Despite this, only a fraction of trajectories can be localized highlighting the difficulty of estimating pose from monocular video. In contrast, the same procedure fails to localize the camera-wearer in Ego4D videos due to rapid head motions and motion blur. On MP3D, pose is available directly from the simulator.

**Performance with ground-truth pose.** In Fig. 8, we investigate the role of pose information for ROOMPRED (top) and NLQ (bottom), by directly embedding ground-truth pose as part of the input to the baseline and our model. We find that our model can benefit from pose on MP3D, but falls short on HouseTours, due to noise in extracted pose (compared to simulator-provided pose in MP3D). However, our approach (with and without pose) outperforms EPC, which explicitly leverages pose both at train and test time.

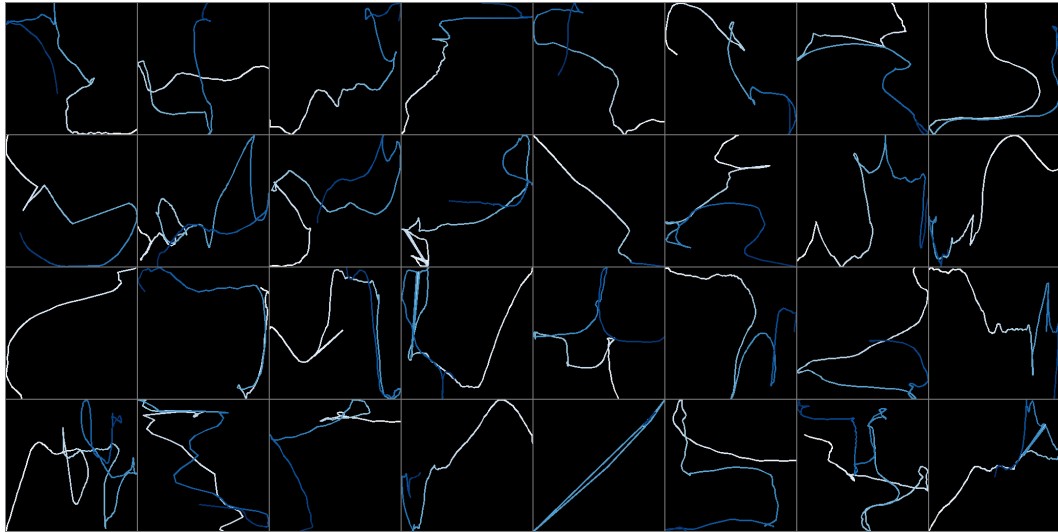

**Figure 9: Camera-pose for HouseTours from COLMAP.** The blue gradient represents the trajectory from start (white) to end (blue).

| | Ego4D [26] | | |
|---|---|---|---|
| RANK1@M → | @0.3 | @0.5 | AVG |
| VSLNET [94] | 5.45 | 3.12 | 4.29 |
| + EGOENV | **6.04** | **3.51** | **4.77** |
| EGOVLP [49] | **10.53** | 5.96 | 8.25 |
| + EGOENV | 10.51 | **6.71** | **8.61** |
| RELER [53] | 10.79 | **6.74** | 8.77 |
| + EGOENV | **11.10** | 6.56 | **8.83** |
| RELER* | 13.68 | 8.23 | 10.96 |
| + EGOENV | **14.40** | **8.54** | **11.47** |
| NAQ [63] | 24.12 | 15.04 | 19.58 |
| + EGOENV | **25.37** | **15.33** | **20.35** |

**Table 9: EgoEnv features with alternate models.** Results on the Ego4D NLQ validation set. RELER* combines EgoVLP [49] features with the model from [53].

### E.3 EgoEnv integrated into other baseline approaches.

In Table 1 in the main paper, we report results on Ego4D using a single architecture and feature combination (VSLNet [94] with SlowFast [22] features). In Table 9 we show results with EgoEnv features integrated into other architectures. Our features consistently improve performance across all architectures, highlighting the complementary environment-level information encoded through our approach.

### E.4 Alternate pretraining task formulations.

In Sec. 3.2.2 in the main paper, we introduced our local state prediction task that is used to pretrain our video encoders on simulated walkthrough videos. We investigate alternate pretraining objectives to validate our task choice. We compare against the following:

- **CARDINALOBJ** is a variant of our local state tasks where we predict only the object categories in each cardinal direction, but not the distances.
- **POSEEMBED** predicts the relative pose (discretized position and orientation) between every pair of observations in a walkthrough video. This is a sub-component of our full model (Sec. 3.2.1).
- **PANOFEAT** directly predicts the image features in each cardinal direction, inspired by prior work on panorama completion [41, 76, 44].

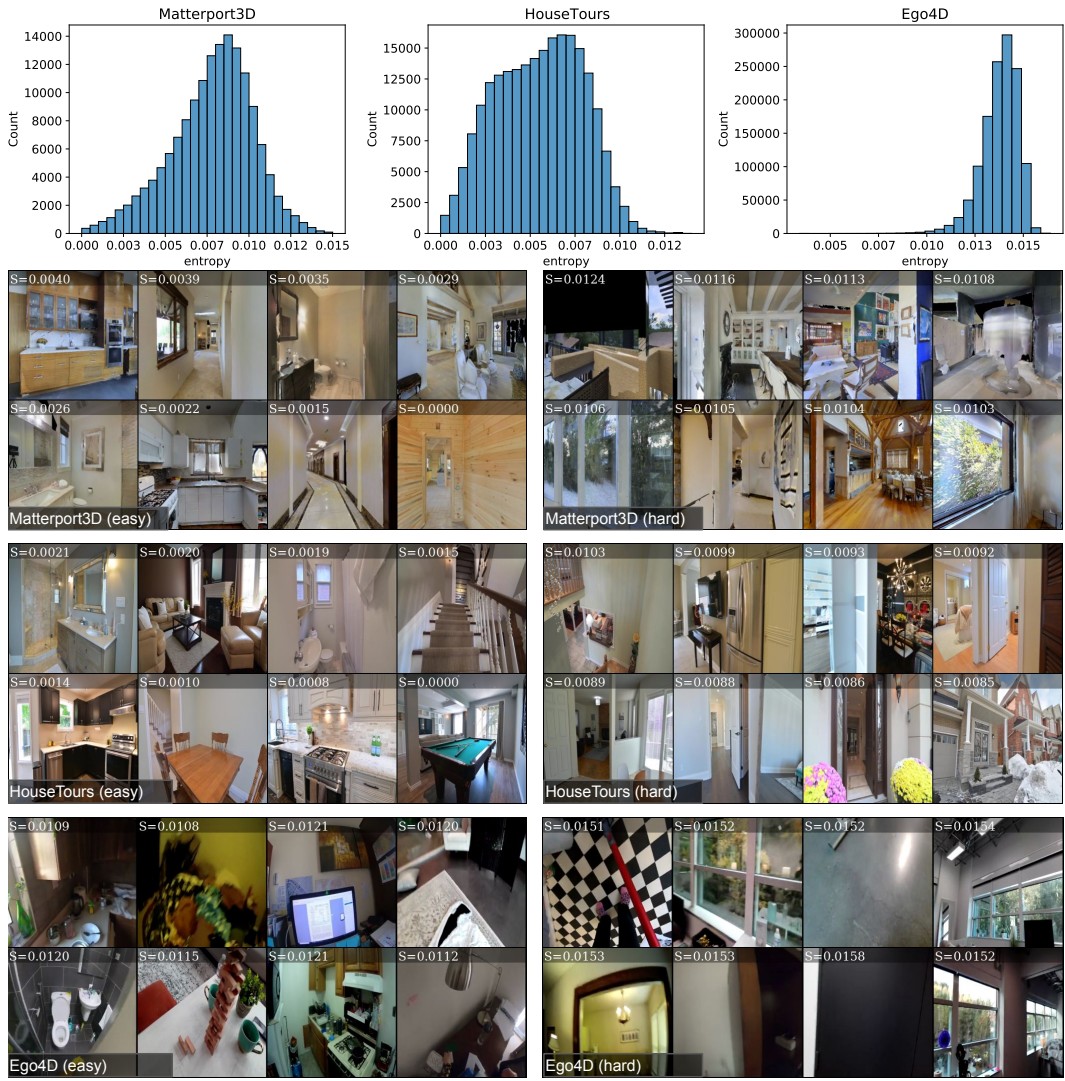

**Figure 10: Illustration of easy vs. hard instances for all datasets. Top panel:** Distribution of entropy scores for ROOMPRED instances. Ego4D instances are skewed towards hard instances due to the egocentric viewpoint and rapid camera and scene motion. **Bottom panel:** Following Sec. 4.2, we sort instances as by their entropy score $S$. We show samples from the top and bottom 10% instances.

- **PANOCONTRAST** uses noise contrastive estimation (NCE) to train a model to predict image features in each cardinal direction. For positives, we use the true image feature in the corresponding direction direction. For negatives, we use image features from the other 3 cardinal directions, as well as trajectory images from different scenes.

Each of these objectives explicitly encodes a combination of semantic and geometric information. For example, PANOFEAT and PANOCONTRAST encodes primarily semantic information as they require reconstruction of image features. POSEEMBED encodes primarily geometric information to predict the relative pose between observation pairs. CARDINALOBJ encodes semantics from object categories and weak geometric information from their relative orientations. Table 10 highlights the performance of these variants on both downstream tasks. Our approach that requires predicting both object labels, orientations as well as rough distances offers a balance of both cues during pretraining, translating to strong downstream performance.

|  | ROOMPRED | | | | NLQ | |
| --- | --- | --- | --- | --- | --- | --- |
|  | MP3D | HT | | | MP3D | HT |
| POSEEMBED | 38.73 | 59.72 | | POSEEMBED | 28.46 | 38.56 |
| CARDINALOBJ | 49.04 | 61.32 | | CARDINALOBJ | 29.32 | 39.34 |
| PANOFEAT | 48.19 | 62.78 | | PANOFEAT | 31.34 | 38.06 |
| PANOCONTRAST | 47.28 | **62.97** | | PANOCONTRAST | **33.22** | 38.56 |
| OURS | **50.40** | 62.68 | | OURS | 32.51 | **43.08** |

**Table 10: Alternate pretraining tasks.** Our local state prediction task offers a good balance of semantic and geometric cues that lead to features with strong downstream performance. For ROOMPRED (left) we report accuracy (%). For NLQ (right) we report mean Rank1@(0.3, 0.5).

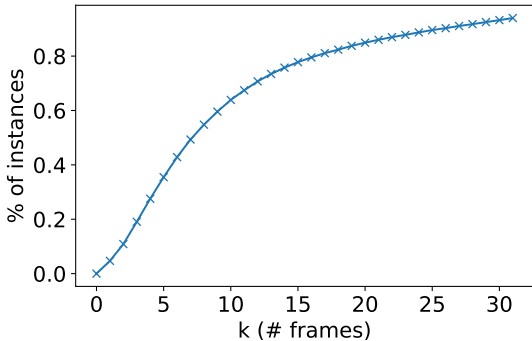

**Figure 11: Percentage of training instances that involve "rare" objects.** The x-axis sets a threshold for what is considered rare. For example, 5% of training instances involve anticipating completely unseen object instances ($k < 1$).

## E.5 Memory vs. anticipation during pretraining

As mentioned in Sec. 3.2 of the main paper, our pretraining task involves elements of both aggregating information about relevant views spread across the walkthrough, as well as anticipating objects that are rarely (or never seen). We quantify this statement in Fig. 11 where we show the percentage of training instances where objects are rarely seen, for different definitions of rarity. For example, 23% of training instances involve predicting objects that appear in only $k < 4$ frames. 5% of training instances involve anticipating completely unseen object instances ($k < 1$).

## E.6 Task-specific pre-training in simulation

As mentioned in Sec. 3.3 of the main paper, our goal is to train task-agnostic representations using videos from simulated agents. The end result is a single model that can generate features for multiple tasks (in our experiments, ROOMPRED and NLQ). This is different from traditional sim-to-real approaches where a new dataset needs to be collected for every downstream task, and a separate model has to be trained on it. Such a dataset needs to be well balanced and carefully designed to match the downstream task. Moreover, as tasks are added, new datasets per task need to be created which may be impractical, especially when they require data beyond the simulator's capability (e.g., simulating human motion, hand-object interaction).

To investigate further, for the ROOMPRED task, we generate a dataset in simulation that maps trajectory frames to room labels[2]. The room categories are estimated directly from the simulator by matching each frame to the nearest navigable point in an annotated room region. We train a ResNet18 model to predict the room category (a six-way classification) and then use features from this model following baselines in Sec. 4 (baselines). The new baseline benefits from representations learned for the *same task* — room prediction — and on the same volume of simulated training data.

On MP3D this performs better than the PlacesCNN baseline (43.3 vs. 42.4%) but is weaker than our model (50.4%). On HouseTours, it performs worse than PlacesCNN (57.9 vs. 58.2%) and our approach (62.7%). The low performance may be attributed to the small label space (only

---

[2]We use trajectories from Gibson [91] scenes as HM3D region annotations are not provided.

six categories), resulting in features that are not discriminative for the large-scale, diverse data downstream. More generally, the task-specific approach is more susceptible to failures due to the sim-to-real gap, which manifests as a lower performance on real-world video frames from HouseTours.

## F  Ablations experiments and additional visualizations.

We present ablation experiments for various model design choices and additional experiments to supplement the discussion in Sec. 4.5 in the main paper.

### F.1  ROOMPRED results with error bars

In Fig. 6 of the main paper, we report results over three runs, by aggregating predictions across all three runs, and then sorting them by difficulty (Sec. 4.2). In Table 11, we show results with standard error *averaged over the three runs* to highlight the variance across approaches. Environment-centric approaches (EPC, TRF, EGOENV) perform better than other baselines, despite higher variance in accuracy. Our approach is consistently the best amongst these.

|  | MP3D | HouseTours | Ego4D |
|---|---|---|---|
| PLACESCNN | $42.39 \pm 0.15$ | $58.24 \pm 0.02$ | $49.50 \pm 0.20$ |
| FRAMEFEAT | $42.04 \pm 0.08$ | $58.70 \pm 0.10$ | $49.34 \pm 0.12$ |
| OBJFEAT | $43.72 \pm 0.06$ | $59.02 \pm 0.12$ | $48.74 \pm 0.11$ |
| MAE | $42.79 \pm 0.25$ | $58.30 \pm 0.08$ | $48.87 \pm 0.08$ |
| EGOTOPO | $41.19 \pm 0.57$ | $58.05 \pm 0.15$ | $49.42 \pm 0.05$ |
| EPC | $42.48 \pm 1.12$ | $61.02 \pm 0.20$ | $-$ |
| TRF (SCRATCH) | $43.27 \pm 0.40$ | $62.12 \pm 0.14$ | $49.65 \pm 0.64$ |
| EGOENV | $\mathbf{50.40 \pm 1.29}$ | $\mathbf{62.68 \pm 0.19}$ | $\mathbf{51.07 \pm 0.65}$ |

**Table 11: ROOMPRED results with error bars across three training runs.**

### F.2  Ablation studies.

We perform ablation experiments for several model design choices listed in Sec. 4 (experiment setup) of the main paper. All ablation experiments are performed on on validation data splits of MP3D and HouseTours.

**Window size $W$.** The window size controls the density of sampled frames for building our environment memory. Larger windows imply temporally separated frame inputs. Our results are in Fig. 12 (left column). We find that $W = 64$ is sufficient for localizing the room category for ROOMPRED, while a $W = 256$ is best for NLQ which requires reasoning over longer horizons.

**Memory size $K$.** The memory size controls the number of frames sampled from a window of size $W$ for building our environment memory. Our results in Fig. 12 (middle column) show the sensitivity of our model to this parameter. On both tasks and datasets, we see only marginal improvements with higher memory sizes, though $K = 64$ results in the best performance.

**Loss weight term $\lambda$.** $\lambda$ controls the weighting between the object and distance prediction term in the local state prediction loss in Sec. 3.2.2. Our results in Fig. 12 (right column) show that $\lambda = 0.1$ results in the best balance between the two semantic and geometric terms.

### F.3  Additional attention visualizations

We present additional examples visualizing the learned attention values in our transformer decoder model in Fig. 13 to supplement Fig. 5 of the main paper. Our model learns to attend to diverse views that are not simply based on temporal adjacency or visual overlap — they capture the surroundings of the camera-wearer.

### F.4  Easy vs. Hard instances in the ROOMPRED task

As mentioned in Sec. 4.2 of the main paper, we sort instances for evaluation by difficulty based on the prediction entropy of a pre-trained scene classifier model. In Fig. 10 (top), we show the distribution

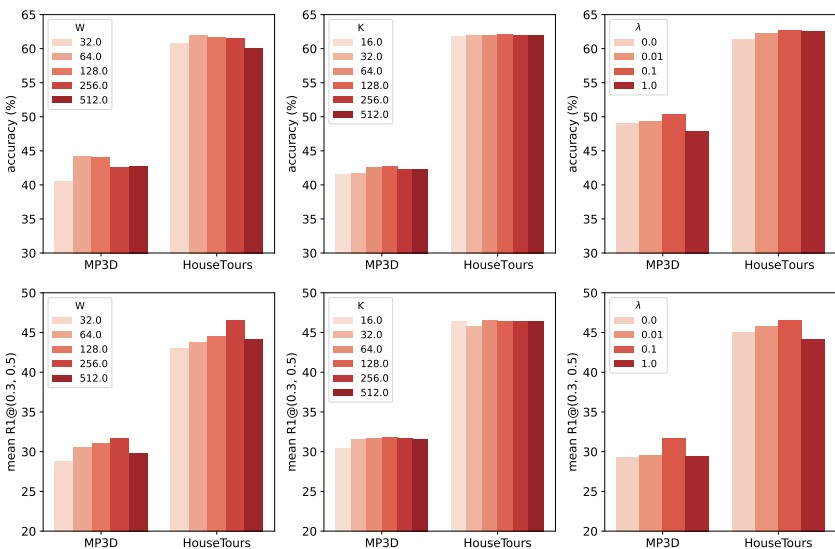

**Figure 12: Ablation experiments on ROOMPRED (top) and NLQ (bottom).** We ablate model hyperparameters: window size $W$ (left), memory size $K$ (middle) and loss weight term $\lambda$ (right). See text for analysis.

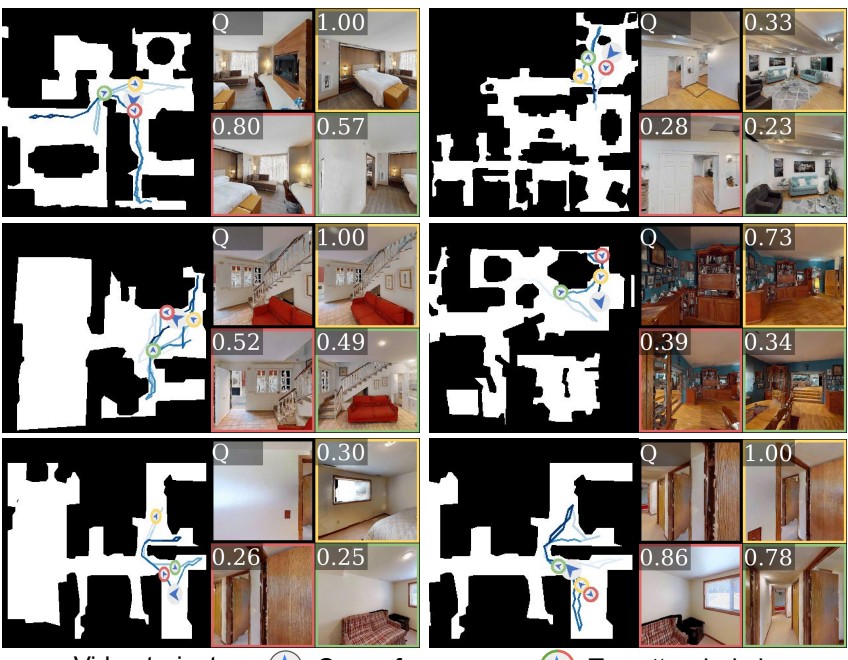

▭ Video trajectory  Ⓐ Query frame pose  Ⓐ Top attended views

**Figure 13: Visualized attention weights.** Following Fig. 5, the query frame (top left) and top-3 attended views (colored boxes), their positions along the trajectory (colored circles), and their associated attention scores are shown.

of this entropy score across all three datasets. In general, Ego4D contains the hardest instances due to characteristic egocentric motion patterns, while HouseTours contains easier examples where the camera-wearer tends to dwell in one particular location to showcase it. We show examples of easy vs. hard instances in Fig. 10 (bottom). Note that the figure only shows the center frame of the clip that is used to predict the room label to highlight the difference between easy and hard frames. Our results in Fig. 6 highlight the advantage of our approach on these hard instances where environment-level reasoning is essential. See our video in Sec. A for more context.