# OpenReview forum: "EgoEnv: Human-centric environment representations from egocentric video"
_NeurIPS.cc/2023/Conference — NeurIPS 2023 oral_

### Official Review · Reviewer_chrV · 2023-07-02

**Soundness:** 3 good
**Presentation:** 3 good
**Contribution:** 3 good
**Rating:** 7
**Confidence:** 4

**Summary:**

This work aims to learn human centric environment representations from first person camera views. Their novel approach utilizes a transformer-based approach that encodes the local environment state at each time-step in an egocentric video is defined as a set of objects, along with their approximate distances, located in front, left, right, and behind the camera-wearer. This work claims to outperform state-of-the-art representations in predicting visited rooms and retrieving significant moments when responding to natural language queries.


**Strengths:**

`Significance:` This work takes an important step towards proposing an approach that models the physical surroundings of a camera from a single egocentric perspective. This work could have multiple applications in AR, VR and robot navigation.

`Originality:` Even though there has been a lot of work in the domain of video understanding in 3D environments, most approaches localize the camera-wearer but do not learn representations for the camera-wearer’s surroundings. Therefore, to the best of my knowledge, this work is novel.

`Quality:` The authors provide a good theoretical background to their approach and back up their claims with rigorous experimentation on multiple datasets and tasks.

`Clarity:` The paper and the supplementary materials are clearly written and easy to follow.

**Weaknesses:**

- This work does not consider the influence of motion blur if the person wearing the camera makes sudden movements, making it challenging to apply this work to real environments.
- This work also uses a semantic segmentation model to identify physical objects in the camera wearers’s environment. However, that could be limiting as their approach is restricted to objects that are classified by the segmentation model.


**Questions:**

- The authors should elaborate on how their models would deal with motion blur and sudden movements of the camera. Is there a motion model that is implicitly learned based on the camera movements or is it assumed that the camera moves at a fixed speed?
- Since this method is intended for AR applications, were their any evaluations done to benchmark the real-time inference performance of this method?


**Limitations:**

The authors do not explicitly state the limitations of this work, and should refer to the Weaknesses section of the review and consider stating those limitations in the paper (if applicable).

---

> ### Author Rebuttal · Authors · 2023-08-09
>
> > W1. This work does not consider the influence of motion blur if the person wearing the camera makes sudden movements, making it challenging to apply this work to real environments.
> > Q1. The authors should elaborate on how their models would deal with motion blur and sudden movements of the camera. Is there a motion model that is implicitly learned based on the camera movements or is it assumed that the camera moves at a fixed speed?
>
> As mentioned in L151, the pose embedding model was intentionally designed to account for missing or unreliable pose estimates in real-world video due to rapid camera-motion and motion blur, however, there is no explicit mechanism in place to filter them out. In terms of inputs to the model, models are trained assuming a fixed speed (i.e., a fixed FPS for the simulated agent trajectories). One of our testbeds, Ego4D, exhibits frequent motion blur; please see the Supp video for examples.
>
> ---
> > W2. This work also uses a semantic segmentation model to identify physical objects in the camera wearers’s environment. However, that could be limiting as their approach is restricted to objects that are classified by the segmentation model.
>
> Yes, this is an important point. As mentioned in L231, we are limited to the 23 categories present in HM3D, detected by semantic segmentation models; however, these categories capture a broad range of commonly available household objects (TV sets, couches, tools etc.) in our video datasets (Ego4D, HouseTours). Importantly, the object classes are *not* a required output of the model – they are used only as labels in the local state task to train the model to link views across a trajectory. Once trained, the prediction heads are discarded while the encoder is retained to generate the environment feature $h_q$ (L192). In short, a larger set of objects is desirable for training, but does not restrict our approach for generating downstream environment features.
>
> ---
> > Q2. Since this method is intended for AR applications, were their any evaluations done to benchmark the real-time inference performance of this method?
>
> No, real-time inference is an important feature but was not a research focus in this work.
>
> ---
> > L1. The authors do not explicitly state the limitations of this work, and should refer to the Weaknesses section of the review and consider stating those limitations in the paper (if applicable).
>
> Please see the discussion on limitations in the common response.

---

> > ### Comment · Reviewer_chrV · 2023-08-11
> >
> > I have read the rebuttal, and am satisfied with the answers to my concerns. Therefore, I am increasing my rating from Weak Accept to Accept.

---

### Official Review · Reviewer_eEwQ · 2023-07-06

**Soundness:** 3 good
**Presentation:** 3 good
**Contribution:** 3 good
**Rating:** 7
**Confidence:** 3

**Summary:**

In this paper, the authors address the limitation of current video understanding methods that only analyze short video clips in isolation, without considering the broader context of the camera-wearer's environment. They propose an approach that establishes a connection between egocentric videos and the surrounding environment by learning predictive representations. To accomplish this, the authors train their models using videos captured by agents in simulated 3D environments, where the environment is fully observable. An interesting finding is that despite being exclusively trained on simulated videos, the proposed approach effectively handles real-world videos from HouseTours and Ego4D datasets. It also achieves state-of-the-art results in the Ego4D NLQ challenge.

I have increased the recommendation after the rebuttal. The rebuttal addressed my concerns.



**Strengths:**

Overall, this paper introduces an innovative approach that bridges the gap between egocentric video and the camera-wearer's environment. By leveraging predictive representations, trained on simulated videos, the proposed method demonstrates improved performance over traditional clip-based approaches in various human-centric video tasks and real-world scenarios. The general idea of grounding egocentric video in its underlying world environment is very interesting. The proposed method to learn representations that are predictive of their surroundings and then enhance standard clip-based models is technically sound.



**Weaknesses:**

In my opinion, the core of this method is a pertaining process that refines a feature into a better one by implicitly seeing its surroundings.
In this sense, in all the experiments, EgoEnv features have knowledge about the surroundings however other features do not. Thus, one important experiment needed to be done is allowing other models to access the same amount of information but in a more straightforward manner.

For example, giving other models not only the current clip as input, but also frames that are uniformly sampled from the whole video.

From the Tables, FrameFeat and ObjectFeat baselines are already very useful. I wonder whether a simple extension of these methods that, similarly, allows a longer temporal range as input at a time, would also perform reasonably well.

**Questions:**

1. The experiments stated in the weakness section.
2. Per the NeurIPS requirement, I hope the authors can discuss the limitations of the paper.

**Limitations:**

The authors did not state the limitations of this work in the paper.

---

> ### Author Rebuttal · Authors · 2023-08-09
>
> > W1/Q1. In my opinion, the core of this method is a pertaining process that refines a feature into a better one by implicitly seeing its surroundings. In this sense, in all the experiments, EgoEnv features have knowledge about the surroundings however other features do not. Thus, one important experiment needed to be done is allowing other models to access the same amount of information but in a more straightforward manner.
>
> Please note that we do provide such baselines.  As mentioned in L294, all models have access to the full video, while inference is performed at a particular time-step. Several baselines do already use the entire video context. TRF (scratch) and EPC both use the same inputs as our model (uniformly sampled across the video) and Ego-Topo (which builds a graph over the entire video’s frames). TRF (scratch) is in fact a combination of FrameFeat + the transformer to allow longer temporal range.
>
> ---
> > Q2. Per the NeurIPS requirement, I hope the authors can discuss the limitations of the paper.
> > The authors did not state the limitations of this work in the paper.
>
> Please see the discussion on limitations in the common response.

---

### Official Review · Reviewer_ssfq · 2023-07-06

**Soundness:** 4 excellent
**Presentation:** 3 good
**Contribution:** 3 good
**Rating:** 8
**Confidence:** 3

**Summary:**

The paper proposes a novel framework to learn environment-aware video representations from egocentric videos. The framework can be trained on synthetic data and incorporated into various existing approaches for real-world downstream tasks including RoomPred and NLQ. Experiments demonstrate that models equipped with the framework achieve superior performances in the two downstream tasks on two real-world egocentric datasets, which demonstrates the value of synthetic data for real-world 3D understanding under egocentric videos.

**Strengths:**

(1) The method in the paper releases the need for camera poses, making it more robust to the noise in structure-from-motion algorithms compared to existing pose-based methods.

(2) Experiments fully demonstrate the effectiveness of the method.

(3) Comprehensive ablation studies demonstrate the value of the core designs of the method.

(4) The method is trained on only automatically-generated synthetic data yet performs better than baselines on real-world scenarios, indicating that simulated data could directly benefit real-world egocentric video understanding without techniques of sim-to-real transfer.

(5) The writing is clear and well-organized.

**Weaknesses:**

(1) The method hugely relies on the surrounding background objects in the 3D scene. When a person navigates in a scene with clean backgrounds, it might be challenging for the method to encode expressive environment features. When dealing with easy real-world instances in the RoomPred task, the method performance is slightly worse than the baselines.

(2) The performance gain from the proposed method seems marginal on Ego4D.

**Questions:**

When incorporating the ground-truth camera poses into the method, I am curious about how to input the pose to EgoEnv+pose since I do not find a network parameter that transfers camera poses to pose embeddings.

**Limitations:**

One potential limitation is that the method has not been tested on dynamic scenes with dynamic background objects or humans such as multi-person interaction scenarios.

---

> ### Author Rebuttal · Authors · 2023-08-09
>
> > W1 The method hugely relies on the surrounding background objects in the 3D scene. When a person navigates in a scene with clean backgrounds, it might be challenging for the method to encode expressive environment features. When dealing with easy real-world instances in the RoomPred task, the method performance is slightly worse than the baselines.
>
> Yes, as mentioned in L187 our method builds on learned priors of objects and their layouts and will be equivalent to standard video models in, for example, an empty room. However, empty scenes may not be aligned with the practical relevance of our models (i.e., assistive robots / AR mentioned in L58-62) where objects and cluttered scenes are key.
>
> ---
> > W2 The performance gain from the proposed method seems marginal on Ego4D.
>
> We present a detailed analysis of our approach’s performance on Ego4D in Supp. B. To summarize, Ego4D videos are in-the-wild videos of natural human activity in diverse scenes. This is in contrast to the simulated walkthroughs used for pretraining. We show that our approach performs well on samples that are aligned with the pretraining data – i.e., indoor home scenarios and navigation heavy scenarios, with lower improvements on out-of-distribution scenes like outdoor activities (e.g., golfing, outdoor cooking). Importantly, while the performance improvement is indeed lower than improvements on HouseTours, our method was the top-ranked approach on Ego4D for NLQ at the time of submission, and remains third ranked on the NLQ leaderboard, demonstrating its impact.
>
> ---
> > Q1 When incorporating the ground-truth camera poses into the method, I am curious about how to input the pose to EgoEnv+pose since I do not find a network parameter that transfers camera poses to pose embeddings.
>
> The input poses are directly transformed to the dimension of the pose embeddings ($p_t$ in Fig 3) using a linear layer.
>
> ---
> > L1. One potential limitation is that the method has not been tested on dynamic scenes with dynamic background objects or humans such as multi-person interaction scenarios.
>
> For testing, Ego4D contains videos with dynamic objects, object interactions, and social interactions. For training, yes, these are interesting future directions that will be driven by advances in simulator capabilities.

---

> > ### Comment · Reviewer_ssfq · 2023-08-19
> >
> > Thanks to the author's response. My concerns have been fully addressed. I believe the work is a solid contribution to the community and would like to maintain my initial rating.

---

### Official Review · Reviewer_9ZdK · 2023-07-07

**Soundness:** 4 excellent
**Presentation:** 3 good
**Contribution:** 4 excellent
**Rating:** 8
**Confidence:** 3

**Summary:**

In the paper, a model is introduced to extract vector embeddings of environments using images from a first-person perspective. The model is trained in a simulated environment where a virtual agent moves around and collects images to learn about its surroundings. The model's performance was tested in two different tasks - predicting the layout of a room and remembering sequences of events. The experimental results showed that the proposed model outperforms other baselines in both downstream tasks.

**Strengths:**

- The paper is well-structured and easy to follow, with a clear motivation behind it. The tables and figures are helpful in explaining the concepts. Overall, I found the paper to be an enjoyable read.

- The paper tackles the very interesting and relevant problem of scene understanding from first-person images.

- The proposed model is very interesting and the experiments are enlightening and helpful in utilizing environment information of egocentric images to better understand the scene.

- I believe this is a solid paper that deserves to be communicated.

**Weaknesses:**

Although the paper is well-written and structured, many important experiments and discussions were excluded and can only be found in the 18-page supplemental material. One example is the full ablation procedure which is solely available in the supplemental material.

I couldn't find any information on the limitations of the agent, which is crucial in this work. For instance, how well the agent performs in outdoor environments? Moreover, it's important to know and if there are any differences in walking patterns between the virtual agent and real people that may affect the results. Additionally, it would be helpful to know how effective the vector embedding is in dynamic scenes where the environment is constantly changing during video acquisition.

I believe that it would be beneficial to briefly discuss these questions that have been raised.


**Questions:**

1) How was the algorithm the generate the walking pattern of the virtual agent? Were used different speeds or motions (e.g., running, walking, etc.) during the training?

2) Based on the images presented in the paper, it appears that the agent did not look up or down. I am curious if this lack of movement would impact its performance in videos focused on activities such as cooking, where the camera angle is often directed downwards. Has there been any analysis on this topic?


**Limitations:**

I was unable to locate the discussion on limitations either in the paper or the lengthy supplemental material.

---

> ### Author Rebuttal · Authors · 2023-08-09
>
> > W1. Although the paper is well-written and structured, many important experiments and discussions were excluded and can only be found in the 18-page supplemental material. One example is the full ablation procedure which is solely available in the supplemental material.
>
> We tried to prioritize the most important information in the main paper, but of course, some experiments had to be moved to Supp. If our paper is accepted, we will move relevant sections to the main paper. For example, the discussion on the limitations of our approach and the ablation experiments mentioned. In the main paper, we point to specific experiments in the supplementary so that a reader knows what is available.
>
> ---
> > W2. I couldn't find any information on the limitations of the agent, which is crucial in this work.
>
> Please see the common response for a discussion on the limitations. Below are responses to specific questions.
>
> > “how well the agent performs in outdoor environments?”
>
> As mentioned in L220-3 and Supp. B, our approach is best suited to indoor environments that are aligned with training scenes. Supp. Table 1 quantifies this for the NLQ task on Ego4D, where we see the largest improvements in indoor home scenarios (e.g., listening to music, household management) and navigation heavy scenarios (walking indoors and outdoors) and lower improvements in outdoor scenes (e.g., golfing, outdoor cooking). Note that our approach is compatible with outdoor video; however, it is limited by the availability of simulated data for activities in outdoor scenes.
>
> > Additionally, it would be helpful to know how effective the vector embedding is in dynamic scenes where the environment is constantly changing during video acquisition.
>
> Interesting point. Our walkthroughs are generated in static scenes where objects are not moved, while the real world is dynamic. We argue in Supp. C1 that a large part of real-world environments are static (e.g., counter tops, staircases; beds, couches, TV sets) which is valuable to encode even when some objects may have moved around. Our embeddings do perform better where there is primarily camera motion and low scene motion / object interaction – Supp. Table 1 shows increased performance in navigation-heavy videos (e.g., walking indoors and outdoors). Further, the larger improvements across both tasks in HouseTours, where the environment is also static, as compared to Ego4D, also hints at this effect.
>
> ---
> > Q1-2. Moreover, it's important to know and if there are any differences in walking patterns between the virtual agent and real people that may affect the results.
> > How was the algorithm the generate the walking pattern of the virtual agent? Were used different speeds or motions (e.g., running, walking, etc.) during the training? Based on the images presented in the paper, it appears that the agent did not look up or down. I am curious if this lack of movement would impact its performance in videos focused on activities such as cooking, where the camera angle is often directed downwards. Has there been any analysis on this topic?
>
> Details about the simulated agent walkthroughs are in Supp. C1. To summarize, agents follow the shortest-path between two points for a fixed number of steps. The action space is discrete (move forward by 0.25m, turn left/right). The frames were rendered into videos using a fixed FPS (i.e., equivalent to a single speed of walking). The episode length and FPS were selected to approximately match the characteristics of human walkthroughs in HouseTours (i.e., agents move between approximately 20 rooms on average in an episode). We did not experiment with agents looking up/down or in general more “human-like” head motion or navigation policies, though that is an interesting direction for future research.

---

> > ### Comment · Reviewer_9ZdK · 2023-08-18
> >
> > After reading the rebuttal, I remain confident in my original assessment and will maintain my initial score. I believe this is a solid paper that deserves to be communicated.

---

### Official Review · Reviewer_iTsH · 2023-07-10

**Soundness:** 3 good
**Presentation:** 4 excellent
**Contribution:** 3 good
**Rating:** 7
**Confidence:** 4

**Summary:**

The work presents an approach to learn spatial environment representations for egocentric videos. Such representations encode the camera-wearer’s (seen and unseen) local surroundings/environment. Previous approaches mostly focus on learning representations over a longer temporal space, however, understanding of the physical spatial space seem to be missing which is addressed in the work via learning of environment-aware features.

The method define a local environment state having a set of object in a relative direction to the camera wearer. The local state has both geometric information (relative object location) and semantic information (object labels). The model learns this local state by learning two matrices - a direction matrix which represents which object is in front, back, left or right and another metric which represents the distance of the object from the camera wearer in a discretized space. To define the matrices, pose information is needed. Since ground truth pose information is missing from real-world egocentric videos, a model first learns the pose information from simulated environment by minimizing a cross entropy loss between the pose of camera wearer and pose of object.

Next, an environment memory is encoded from a video walkthrough of T frames and a query frame. Pose embeddings are generated for all the T frames and query frame using the learned pose model, and each frame is encoded with this pose embedding. Then, K video frames are sampled to construct an environment memory using a transformer encoder. This representation consists of both temporal and spatial information of the environment. The transformer decoder then uses this environment memory and the query frame to generate a EgoEnv representation which is combined with the original video features (eg. Resnet). These representations are finally used for the video downstream tasks.

The method is evaluated on the room prediction and NLQ challenge on the datasets - Ego4D, HouseTours, and Matterport3D.

**Strengths:**

1. The paper is well-written and has a detailed supplementary material covering the dataset statistics, model architecture details, and ablation experiments.
2. The work shows a thorough evaluation with the method being evaluated on multiple datasets and compared with multiple baselines.

**Weaknesses:**

1. It would be great to discuss about the accuracy of the pose embedding learning network on the simulated network since the overall model is dependent on it.
2. There can be more discussion as to how the sim-to-real gap gets reduced in the approach.

**Questions:**

How much time does it take to train the pose embedding network and the rest of the model too in general?

**Limitations:**

No such limitations. The paper is well-written, covers all the implementation details, and discusses about all the aspects of the approach.

---

> ### Author Rebuttal · Authors · 2023-08-09
>
> > W1 It would be great to discuss about the accuracy of the pose embedding learning network on the simulated network since the overall model is dependent on it.
>
> Thanks for the suggestion. As mentioned in L157, the pose embedding network is trained to predict relative pose discretized into 12 angles and 4 distance ranges. On the validation set, the model achieves accuracies of 48.4% on relative distance prediction and 34.4% on relative orientation prediction. Note that this task is challenging – models must predict relative pose for all possible pairs of observations in a trajectory using their visual features alone – however the goal is to generate pose encodings, not to output perfect pose. In Supp. E.2.1, we highlight the importance of these pose embeddings for local state prediction. In short, including pose embeddings leads to better object class and distance prediction, especially when objects are not immediately visible.
>
> ---
> > W2 There can be more discussion as to how the sim-to-real gap gets reduced in the approach.
>
> We make efforts to minimize the sim-to-real gap in our dataset and task design. First, we opt for photo-realistic environments from HM3D which contain high-fidelity reconstructions of diverse, real-world houses. Second, when creating the simulated trajectories in Sec. 4 (Simulator environments), we try to match the characteristics of simulated agent walkthroughs to camera-wearer movement in video datasets (L233). Specifically, we position cameras at head-level for the simulated agents, and we adjust the episode duration such that the number of rooms visited per episode by simulated agents roughly matches that in HouseTours (approximately 20 room transitions on average). Please see our Supp. video to compare the generated walkthroughs with HouseTours videos. Finally, we design our local state prediction task around capturing object and environment layout that reflects priors of real-world object distributions. We select this over other simpler alternatives like predicting image features directly, which are more susceptible to failures due to sim-to-real visual differences. We compare alternatives in Supp. E.4.
>
> We discuss the effect of the sim-to-real gap in Supp B. To summarize, we find that our approach works best on scenes that are aligned with our simulated training data (i.e., indoor home scenarios; videos with lots of walking and less object interaction) and naturally performs worse on out-of-distribution activities (e.g., golfing, outdoor cooking).
>
> ---
> > Q1. How much time does it take to train the pose embedding network and the rest of the model too in general?
>
> Both the local state prediction and the pose embedding training is performed for 2.5k epochs (L262). Each training run takes ~24 hours when trained on two Quadro RTX 6000 GPUs. Note that this is a one time pre-training cost – once trained, the model is directly used as a feature extractor in downstream tasks and does not incur any extra training cost.

---

> > ### Comment · Reviewer_iTsH · 2023-08-21
> >
> > I have read the author’s rebuttal and my concerns have been fully addressed. This work is a good contribution towards egocentric research and thus, I will maintain my initial rating.

---

### Author Rebuttal · Authors · 2023-08-09

Thanks to all the reviewers for their effort and constructive feedback. All five reviewers recommend accepting the paper, with two recommending strong accept (5, 6, 7, 8, 8). We address common concerns shared by reviewers below.

**Limitations of the proposed approach**

We will emphasize the limitations more in the paper and summarize here.

We discuss the limitations of our approach in the context of the sim-to-real gap in L328 and Supp. B. Specifically, our model is affected by the type and diversity of pretraining data — videos of simulated agents walking around a house — limiting its generalization to unconstrained real-world video. Similarly, our approach is limited by simulator functionality – HM3D scenes support a small set of objects, which may not overlap with real-world environments, and Habitat does not support fine-grained object-interactions (e.g., chopping vegetables). As a result, we find that our approach works well on videos that are consistent with pretraining (i.e., indoor home scenarios; videos with lots of walking and less object interaction) but contributes less on out-of-distribution scenes and activities (e.g., golfing, outdoor cooking). We expect that future advancements in simulator capabilities (e.g., human motion models for agents, fine grained object interaction simulation) will help address this class of limitations.

Beyond the sim-to-real gap, our approach has other limitations. First, our model does not have specialized modules to aggregate long-term temporal (or pose) information into its representation, compared to, for example, structure from motion methods that can aggregate and re-localize observations over a long video. As a result, our approach does not see benefits from increasing the temporal window from which the memory is constructed (Supp. F3). Next, our local state prediction task is learned in a coarse 2D space – the top-down map of the environment – which does not encode fine-grained geometric relations which may be important for certain tasks (e.g., is the object placed on top of, or inside another?). Finally, our approach is computationally intensive. While pre-training is a one-time cost, generating each augmented clip feature at inference requires the computation of several frame features in the vicinity of the clip, and then aggregating that information using the transformer module. This may be limiting for real-time inference applications in the assistance setting.

Despite these limitations, our approach outperforms state-of-the-art representations for predicting visited rooms and retrieving important moments from natural language queries.

---

### Decision · Program_Chairs · 2023-09-21

**Decision:**

Accept (oral)

**Comment:**

Reviewers are unanimously enthusiastic about this work -- noting that the approach to environmental representation learning is well-motivated and trainable in simulation. Much of the discussion during the rebuttal phase clarified limitations of the proposed approach which should be integrated with the current text.